# HOMOMORPHISM COUNTS AS STRUCTURAL ENCODINGS FOR GRAPH LEARNING

**Linus Bao**[*]
University of Oxford

**Emily Jin**[*]
University of Oxford

**Michael Bronstein**
University of Oxford / AITHYRA

**İsmail İlkan Ceylan**
University of Oxford

**Matthias Lanzinger**
TU Wien

## ABSTRACT

Graph Transformers are popular neural networks that extend the well-known Transformer architecture to the graph domain. These architectures operate by applying self-attention on graph nodes and incorporating graph structure through the use of positional encodings (e.g., Laplacian positional encoding) or structural encodings (e.g., random-walk structural encoding). The quality of such encodings is critical, since they provide the necessary *graph inductive biases* to condition the model on graph structure. In this work, we propose *motif structural encoding* (MoSE) as a flexible and powerful structural encoding framework based on counting graph homomorphisms. Theoretically, we compare the expressive power of MoSE to random-walk structural encoding and relate both encodings to the expressive power of standard message passing neural networks. Empirically, we observe that MoSE outperforms other well-known positional and structural encodings across a range of architectures, and it achieves state-of-the-art performance on a widely studied molecular property prediction dataset.

## 1 INTRODUCTION

Graph Neural Networks (GNNs) have been the prominent approach to graph machine learning in the last decade. Most conventional GNNs fall under the framework of Message Passing Neural Networks (MPNNs), which incorporate graph structure – the so-called *graph inductive bias* – in the learning and inference process by iteratively computing node-level representations using "messages" that are passed from neighboring nodes (Gilmer et al., 2017). More recently, Transformer architectures (Vaswani et al., 2017) have been applied to the graph domain and achieve impressive empirical performance (Rampásek et al., 2022; Ma et al., 2023), especially in molecular property prediction.

While MPNNs operate by exchanging messages between adjacent nodes in a graph, Transformers can be seen as a special type of GNN that operates on complete graphs. On the one hand, this allows for direct communication between all node pairs in a graph, regardless of whether there exists an edge between two nodes in the original input graph. On the other hand, since the node adjacency information is omitted, the Transformer lacks any "built-in" graph inductive bias. Instead, the underlying graph structure is usually provided by combining Transformer layers with local message-passing layers (Yun et al., 2019; Rampásek et al., 2022; Bar-Shalom et al., 2024) or by incorporating additional pre-computed features that encode the topological context of each node. These additional features are referred to as *positional or structural encodings*. Common encodings include Laplacian positional encoding (LapPE) (Dwivedi et al., 2023) and random-walk structural encoding (RWSE) (Dwivedi et al., 2022a). The quality of these encodings are a key ingredient in the success of Transformers on graphs (Dwivedi et al., 2022a; Rampásek et al., 2022).

**Motivation.** While empirical studies have observed the impact of structural or positional encodings on model performance (Dwivedi et al., 2022a; Rampásek et al., 2022), our theoretical understanding of the expressive power of different encodings remains limited. This represents an important gap in the literature, especially since the expressive power of

---

[*]Equal contribution.

Transformer-based architectures heavily rely on the specific choice of the structural or positional encoding (Rosenbluth et al., 2024). Let us consider RWSE, which has been empirically reported as the most successful encoding on molecular benchmarks (Rampásek et al., 2022). We now illustrate a serious limitation of RWSE in its power to distinguish nodes.

**How Expressive is RWSE?** The expressive power of MPNNs is upper bounded by the *1-dimensional Weisfeiler Leman graph isomorphism test (1-WL)* (Xu et al., 2019; Morris et al., 2019). This means that the node invariants computed by MPNNs are at most as powerful as the node invariants computed by 1-WL. As a result, an MPNN can distinguish two nodes *only if* 1-WL can distinguish them. Some MPNN architectures, such as Graph Isomorphism Networks (GIN) (Xu et al., 2019), can match this expressiveness bound and distinguish any pair of nodes that can be distinguished by 1-WL. We relate the expressive power of RWSE to that of the Weisfeiler Leman hierarchy by proving that node invariants given by RWSE are strictly weaker than node invariants computed by 2-WL (Proposition 4.4).[1] Additionally, RWSE node invariants are incomparable to node invariants computed by 1-WL (Proposition 4.7). In fact, there are simple node pairs which can be distinguished by 1-WL but not by RWSE (and vice versa).

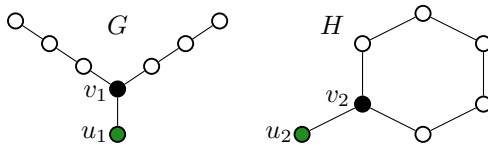

(a) The nodes $u_1$ and $u_2$ (also, $v_1$ and $v_2$) can be distinguished by 1-WL, but not by RWSE.

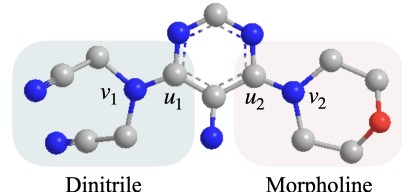

(b) A molecule from the ZINC dataset, with Carbon, Nitrogen, and Oxygen atoms. The nodes $u_1$ and $u_2$ are indistinguishable by RWSE and so are $v_1$ and $v_2$.

Figure 1: RWSE is weaker than 1-WL in distinguishing certain nodes (a), which holds for *all* choices of random walk length. This limitation is observed in real-world molecular graphs (b).

*Example* 1.1. Consider the nodes from the graphs $G$ and $H$ from Figure 1(a). Observe that the nodes $u_1$ and $u_2$ can be distinguished by 1-WL. Interestingly, however, they are indistinguishable to RWSE for *all* lengths $\ell$ of the considered random walk. This observation also applies to the nodes $v_1$ and $v_2$. Moreover, this limitation is not merely of theoretical interest, as it readily applies to real-world molecules, where the use of RWSE is prominent. Figure 1(b) depicts a molecule from the ZINC dataset, where the nodes $u_1$ and $u_2$ (also, $v_1$ and $v_2$) cannot be distinguished by RWSE although they can be distinguished by MPNNs. In practical terms, this implies that RWSE cannot distinguish key nodes in a Dinitrile group from those in a Morpholine group.                          △

**Motif Structural Encoding as a Flexible and Powerful Approach.** In this paper, we propose *motif structural encoding* (MoSE) as a method of leveraging homomorphism count vectors to capture graph structure. Homomorphism counts have been investigated in the context of MPNNs to overcome well-known theoretical limitations in their expressivity (Barceló et al., 2021; Jin et al., 2024) with promising theoretical and empirical findings (see Section 2). Building on the existing literature, we show that MoSE is a general, flexible, and powerful alternative to existing positional and structural encoding schemes in the context of Transformers. In fact, we show that unlike RWSE, MoSE cannot be strictly confined to one particular level of the WL hierarchy (Proposition 4.3), and MoSE can provide significant expressiveness gains exceeding that achieved by RWSE (Theorem 4.6).

*Example* 1.2. Consider the graphs $G$ and $H$ from our running example, and observe that $H$ has a cycle of length six, while $G$ has no cycles. The node-level homomorphism counts $\mathsf{Hom}_{\to v_1}(⬡, G) \neq \mathsf{Hom}_{\to v_2}(⬡, H)$ (defined formally in Section 4.2) provide sufficient information to distinguish the nodes $v_1$ and $v_2$. Analogous statements can also be made for $u_1$ and $u_2$.                          △

**Contributions.** Our key contributions can be summarized as follows:

- We introduce MoSE and detail the expressiveness guarantees that can be achieved by using homomorphism counts as a graph inductive bias (Section 4.1).

---

[1]Throughout the paper, we refer to the folklore version of the WL hierarchy (Grohe & Otto, 2015).

- We compare MoSE to RWSE in terms of expressivity and relate them to the expressive power of MPNNs through the well-known WL hierarchy (Section 4.2).
- We empirically validate our theoretical findings and demonstrate the efficacy of MoSE on a variety of real-world and synthetic benchmarks. We report consistent performance gains over existing encoding types and achieve state-of-the-art results on the ZINC molecular benchmark (Section 5).

## 2 RELATED WORK

**(Graph) Transformers and Encodings.** Vaswani et al. (2017) first highlighted the power of self-attention when they introduced their Transformer architecture. Dwivedi & Bresson (2020) generalized the concepts presented in Vaswani et al. (2017) to the graph domain. In recent years, several different Graph Transformer architectures have arisen. Yun et al. (2019) first combined message-passing layers with Transformer layers, and other models have followed this approach (Rampásek et al., 2022; Bar-Shalom et al., 2024; Shirzad et al., 2023). Relatedly, Transformer architectures dedicated to knowledge graphs have also been developed (Zhang et al., 2023c; Liu et al., 2022). Shehzad et al. (2024) present a survey of existing Graph Transformer models and how they compare to each other.

A key component in the success of Graph Transformer models is the use of effective graph inductive biases (Dwivedi et al., 2022a). Laplacian positional encodings (LapPE), which were introduced by Dwivedi & Bresson (2020), are a popular choice, but they break invariance[2] as they are dependent on an arbitrary choice of eigenvector sign and basis (Lim et al., 2023b;a). Rampásek et al. (2022) present a framework for building general, powerful, and scalable Graph Transformer models, and they perform ablations with a suite of different encoding types, promoting the use of random-walk structural encoding (RWSE) on molecules. Ying et al. (2021) introduced Graphormer, which uses an attention mechanism based on shortest-path (and related) distances in a graph. Ma et al. (2023) present GRIT as a novel Graph Transformer that relies solely on relative random walk probabilities (RRWPs) for its graph inductive bias.

**Expressive power of GNNs.** MPNNs capture the vast majority of GNNs (Gilmer et al., 2017) and their expressive power is upper bounded by 1-WL (Xu et al., 2019; Morris et al., 2019). This limitation has motivated a large body of work, including *higher-order* GNN architectures (Morris et al., 2019; Maron et al., 2019a;b; Keriven & Peyré, 2019), GNNs with unique node features (Loukas, 2020), and GNNs with random features (Sato et al., 2021; Abboud et al., 2021), to achieve higher expressive power. The expressivity of GNNs has also been evaluated in terms of their ability to detect graph components, such as biconnected components (Zhang et al., 2023b), and in terms of universal approximation on permutation invariant functions (Chen et al., 2019). Fully expressive architectures have been designed for special graph classes, e.g., for planar graphs (Dimitrov et al., 2023).

Our work is most closely related to approaches that design more expressive architectures by injecting the counts of certain substructures, widely referred to as motifs. Bouritsas et al. (2023) present an early work that enhances node feature encodings for GNNs by counting subgraph isomorphisms for relevant patterns. Barceló et al. (2021) follow up on this work by proposing to instead count the number of homomorphism from a set of pertinent substructures. Several other works have identified homomorphism counts as a key tool for understanding the expressive power of MPNNs (Neuen, 2023; Lanzinger & Barceló, 2024; Wang & Zhang, 2024). Zhang et al. (2024) also recently proposed a new framework that characterizes the expressiveness of GNNs in terms of homomorphism counts. Through tight connections between counting homomorphisms and counting (induced) subgraphs (Curticapean et al., 2017; Bressan et al., 2023; Roth & Schmitt, 2020), these expressiveness results can be lifted to a wide range of functions (Lanzinger & Barceló, 2024). As such, Jin et al. (2024) built upon these theoretical observations to establish a general framework for using homomorphism counts in MPNNs. In particular, they show that homomorphism counts that capture certain "bases" of graph mappings provide a theoretically well-founded approach to enhance the expressivity of GNNs. Maehara & NT (2024) reiterate the benefits of using homomorphism in GNNs.

---

[2]By sacrificing invariance, it becomes much easier to design very expressive architectures. In fact, Graph Transformer architectures with LapPE are universal (Kreuzer et al., 2021), but so is a 2-layer MLP and a single-layer GNN using LapPE (see Proposition 3.1 of Rosenbluth et al. (2024)).

## 3    PRELIMINARIES

**Graphs and Homomorphism Counts.**    An undirected *graph* is a set of *nodes* $V(G)$ and a set of *edges* $E(G) \subseteq V(G) \times V(G)$ which satisfy symmetry: $(u, v) \in E(G)$ if and only if $(v, u) \in E(G)$. Unless otherwise stated, we take all graphs to be finite, and we take all graphs to be *simple*: $(v, v) \notin E(G)$ for all $v \in V(G)$ and there exists at most one edge (up to edge symmetry) between any pair of nodes. We describe nodes $u, v \in V(G)$ as *adjacent* if $(u, v) \in E(G)$. The set of all nodes adjacent to $v \in V(G)$ is the *neighborhood* of $v$, notated $\mathcal{N}(v)$. The number of neighboring nodes $|\mathcal{N}(v)|$ is the *degree* of $v$, notated $d(v)$.

A *homomorphism* from graph $G$ to graph $H$ is a function $f : V(G) \to V(H)$ such that $(u, v) \in E(G)$ implies $(f(u), f(v)) \in E(H)$. We say a homomorphism is an *isomorphism* if the function $f$ is bijective, and if it additionally satisfies $(u, v) \in E(G)$ if and only if $(f(u), f(v)) \in E(H)$. In this case, we describe the graphs $G$ and $H$ as being *isomorphic*, denoted $G \cong H$. We write $\text{Hom}(G, H)$ for the number of homomorphisms from $G$ to $H$. If we restrict to counting only homomorphisms which map a particular node $g \in V(G)$ to the node $h \in V(H)$, we denote this *rooted homomorphism count* as $\text{Hom}_{g \to h}(G, H)$. Sometimes the choice of $g$ is unimportant, so we use $\text{Hom}_{\to h}(G, H)$ to notate the rooted homomorphism count $\text{Hom}_{g \to h}(G, H)$ where $g$ can be fixed as any arbitrary node in $V(G)$. We use $\text{Hom}(G, \cdot)$ to denote the function which maps any graph $H$ to the integer $\text{Hom}(G, H)$, and we define rooted homomorphism count mappings analogously. If $\mathcal{G}$ is a collection of graphs, then we treat $\text{Hom}(\mathcal{G}, \cdot)$ as a function that maps any input graph $H$ to the ordered[3] tuple (or integer vector) defined as $\text{Hom}(\mathcal{G}, H) = (\text{Hom}(G, H))_{G \in \mathcal{G}}$, and analogously for rooted counts.

Curticapean et al. (2017) describe *graph motif parameters* as functions $\Gamma(\cdot)$ that map graphs into $\mathbb{Q}$, such that there exists a *basis* of graphs $\text{Supp}(\Gamma) = \{F_i\}_{i=1}^{\ell}$ and corresponding coefficients $\{\alpha_i\}_{i=1}^{\ell} \subseteq \mathbb{Q} \backslash \{0\}$ which decompose $\Gamma$ as a finite linear combination $\Gamma(\cdot) = \sum_{i=1}^{\ell} \alpha_i \cdot \text{Hom}(F_i, \cdot)$. Given $G$, the function which maps any $H$ to the number of times that $G$ appears as a subgraph[4] of $H$ is a graph motif parameter (Lovász, 2012) whose basis is known as $\text{Spasm}(G)$. Many mappings of interest can be described as graph motif parameters, so the theory of such bases bolsters homomorphism counts as broadly informative and powerful (Jin et al., 2024).

Nguyen & Maehara (2020) present a generalization of homomorphism counts that allow for vertex weighting. For graph $G$, let $\omega : V(G) \to \mathbb{R}_{\geq 0}$ describe *node weights*. The *weighted homomorphism count* from a graph $F$ into $G$ is given as:

$$\omega\text{-Hom}(F, G) = \sum_{f \in \mathcal{H}} \left( \prod_{v \in V(F)} \omega(f(v)) \right)$$

where $\mathcal{H}$ is the set of all homomorphisms from $F$ to $G$. We define the node-rooted version $\omega\text{-Hom}_{u \to v}(F, G)$ by restricting $\mathcal{H}$ to only those homomorphisms which map $u \in V(F)$ to $v \in V(G)$. The mapping $\omega\text{-Hom}_{\to(\cdot)}(F, \cdot)$ is defined analogously to the un-weighted case: the graph-node pair $(G, v)$ gets mapped to $\omega\text{-Hom}_{u \to v}(F, G)$ where $u \in V(F)$ is fixed arbitrarily. Setting $\omega(v) = 1$ for all $v \in V(G)$ recovers the un-weighted count $\omega\text{-Hom}(F, G) = \text{Hom}(F, G)$, and similarly for the node-rooted version. Weighted homomorphism counts are well-studied in the context of their connection to graph isomorphism (Freedman et al., 2004; Cai & Govorov, 2021), and in the context of the universal approximation capabilities and empirical performance of graph classifiers which use $\omega\text{-Hom}(F, \cdot)$ counts as a graph embedding (Nguyen & Maehara, 2020).

**Expressive Power.**    A *graph invariant* is a function $\xi(\cdot)$ which acts on graphs, satisfying $\xi(G) = \xi(H)$ whenever $G \cong H$ for all graphs $G$ and $H$. For indexed families of graph invariants $\{A_i\}_{i \in I}$ and $\{B_j\}_{j \in J}$, we define the *expressivity* relation $A \preceq B$ to denote: for any choice of $i \in I$, there exists a choice of $j \in J$ such that $B_j(G) = B_j(H)$ implies that $A_i(G) = A_i(H)$ for all $G$ and $H$. If $A \preceq B$ and $B \preceq A$, then we say that $A \simeq B$. We write $A \prec B$ when $B$ has strictly greater expressive power, $A \preceq B$ but $A \not\simeq B$. When we reference $\preceq$ for some fixed graph invariant $\xi(\cdot)$, we interpret $\xi$ as a family which contains only one graph invariant (hence the indexing is trivial).

---

[3]If the graphs in $\mathcal{G}$ are not ordered initially, we arbitrarily fix some indexing of the elements in $\mathcal{G}$ so that they become ordered.

[4]We count the number of times one can find a graph $H'$ with $V(H') \subseteq V(H)$ and $E(H') \subseteq E(H)$ such that $G \cong H'$.

Noting that a GNN architecture can be treated as a family of graph invariants indexed by the model weights, the relation $\preceq$ generalises common notions of the graph-distinguishability expressive power for GNNs (Zhang et al., 2023a). As we will see in Section 4, the $\preceq$ relation naturally extends to expressivity comparisons between structural/positional encoding schemes as well.

**MPNNs and Weisfeiler-Lehman Tests.** Given graph $G$, the 1-WL test induces the node coloring $c^{(t)} : V(G) \to \Sigma$ for all $t$ up until some termination step $T$, at which point the graph-level 1-WL label is defined as the multiset of final node colors 1-WL$(G) = \{\!\{c^{(T)}(v) : v \in V(G)\}\!\}$. Graphs $G$ and $H$ are considered *1-WL indistinguishable* if they have identical graph labels 1-WL$(G) = $ 1-WL$(H)$. Crucially, it holds that MPNN $\preceq$ 1-WL where we treat "MPNN" as a family of graph invariants indexed by both the choice of message-passing architecture and the model parameters, and we treat 1-WL as a singleton family containing only the graph invariant $G \mapsto$ 1-WL$(G)$ (Xu et al., 2019).

We extend the Weisfeiler-Lehman test to "higher dimensions" by coloring $k$-tuples of nodes. The $k$-WL test induces node tuple coloring $c^{(t)} : V(G)^k \to \Sigma$ for $t = 1, ..., T$ iterations, and then it aggregates a final graph-level $k$-WL label analogously to 1-WL (Huang & Villar, 2021). In this work, we refer to the *folklore $k$-WL test*, which is provably equivalent to the alternative *oblivious $k$-WL* formulation up to a re-indexing of $k$ (Grohe & Otto, 2015). It has been shown that the $k$-WL tests form a well-ordered hierarchy of expressive power, where $k$-WL $\prec$ $(k + 1)$-WL (Cai et al., 1989). Here, we treat $k$-WL as a singleton family (trivial indexing) for each particular choice of $k$.

**Self-Attention and Positional/Structural Encoding.** Central to any Transformer architecture is the *self-attention* mechanism. When applied to graphs, a single "head" of self-attention learns a $t^{\text{th}}$-layer hidden representation $\boldsymbol{h}_v^{(t)} \in \mathbb{R}^d$ of every node $v \in V(G)$ by taking the weighted sum $\boldsymbol{h}_v^{(t)} = \phi^{(t)}(\sum_{u \in V(G)} \alpha_{v,u}^{(t)} \boldsymbol{h}_u^{(t-1)})$ where the *attention coefficients* are $\alpha_{v,u}^{(t)} = \psi^{(t)}(\boldsymbol{h}_v^{(t-1)}, \boldsymbol{h}_u^{(t-1)})$. Here, $\phi^{(t)}$ and $\psi^{(t)}$ are learnably-parameterized functions. The most common implementation of $\phi^{(t)}$ and $\psi^{(t)}$ is *scaled dot product attention*, as defined by Vaswani et al. (2017); although some Transformers such as GRIT (Ma et al., 2023) deviate from this. In most Transformers, we utilize multiple attention heads in parallel (whose outputs are concatenated at each layer), and we interweave self-attention layers with fully-connected feed-forward layers.

Since basic self-attention does not receive the adjacency matrix as an input, many Graph Transformer models choose to inject a graph's structural information into the model inputs by way of node positional/structural encoding. For example, the widely used *$k$-length random walk structural encoding* (RWSE$_k$) assigns to each node its corresponding diagonal entry of the degree normalized adjacency matrix[5] $(\boldsymbol{D}^{-1}\boldsymbol{A})^i$ for all powers $i = 1, ..., k$ (Rampásek et al., 2022; Dwivedi et al., 2022a; Ma et al., 2023). Each node's RWSE$_k$ vector is then concatenated or added to the node's initial feature vector,[6] and the resulting node embedding is passed into the Transformer as an input. More broadly, we can define *node positional or structural encoding* to mean any isomorphism-invariant mapping $pe$ which takes in a node-graph pair $(v, G)$ and outputs a vector label $pe(v, G) \in \mathbb{R}^d$. Then, the graph-level PE label is defined as the multiset of node-level labels PE$(G) = \{\!\{pe(v, G) : v \in V(G)\}\!\}$. In this way, we can treat RWSE$_k$ as a family of graph invariants (mapping into multisets with elements in $\mathbb{R}^d$) indexed by $k$. Hence, we can compare the graph-level expressivity of RWSE$_k$ and any other general positional/structural encoding scheme PE under $\preceq$.

## 4  HOMOMORPHISM COUNTS AS A GRAPH INDUCTIVE BIAS

In this section, we formally define motif structural encoding (MoSE). Just as with other node-level structural or positional encodings, MoSE provides a way to encode the structural characteristics of a node in terms of a numerical vector. Such encodings then provide graph inductive bias to models, such as Transformers, that cannot (or only in a limited fashion) take graph structure into account. All proof details are provided in Appendix A.

---

[5]Here, $\boldsymbol{D}$ notates the diagonal degree matrix, and $\boldsymbol{A}$ the adjacency matrix.
[6]Sometimes, the initial node feature and the RWSE vector are passed through MLPs before being combined and fed into the GNN.

## 4.1 MOTIF STRUCTURAL ENCODING (MOSE)

The *motif structural encoding* (MoSE) scheme is parameterized by a choice of a finite[7] pattern graph family $\mathcal{G} = \{G_1, \ldots, G_d\}$, as well as a choice of node weighting scheme $\omega$ which sends any $(v, H)$ to a non-negative weight $\omega(v, H) \in \mathbb{R}_{\geq 0}$. For each node $v$ of graph $H$, the node-level $\text{MoSE}_{\mathcal{G},\omega}$ label of $v$ is:

$$\text{MoSE}_{\mathcal{G},\omega}(v, H) = \left[\omega\text{-Hom}_{\to v}(G_i, H)\right]_{i=1}^{d} \in \mathbb{R}^d \tag{1}$$

We will notate $e_v^{\mathcal{G}} := \text{MoSE}_{\mathcal{G},\omega}(v, H)$ for shorthand when $\omega$ and $H$ are either clear from context, or any arbitrary choice can be made. Unless otherwise specified, we use the constant $\omega$ weighting which sends all nodes in all graphs to 1, aligning $\text{MoSE}_{\mathcal{G},\omega}$ with the un-weighted homomorphism counts used in previous works (Jin et al., 2024; Barceló et al., 2021). In this case, we omit $\omega$ from our notation, using $\text{MoSE}_{\mathcal{G}}$ to denote our encoding. When we reference $\text{MoSE}_{\mathcal{G},\omega}$ at the graph-level, we mean the multiset of node-labels $\text{MoSE}_{\mathcal{G},\omega}(H) = \{\!\{e_v^{\mathcal{G}} : v \in V(H)\}\!\}$. It holds that $\text{MoSE}_{\mathcal{G},\omega}(\cdot)$ is a graph invariant because for any two graphs $H$ and $F$ with isomorphism $\iota : V(H) \to V(F)$, we have $e_v^{\mathcal{G}} = e_{\iota(v)}^{\mathcal{G}}$ for every vertex $v \in V(H)$ (Nguyen & Maehara, 2020).

MoSE offers a highly general and flexible approach to structural encoding on account of two key parameters: the graphs $\mathcal{G}$ and the vertex weight function $\omega$. Both of these parameters can be adapted to fit the problem domain and the model architecture as desired. The choice of $\mathcal{G}$ can be informed in precise terms by desired levels of expressiveness as shown below in Proposition 4.3. Furthermore, the choice of $\mathcal{G}$ can build on a range of empirical studies on structural information in MPNNs (Barceló et al., 2021; Jin et al., 2024; Wang & Zhang, 2024; Bouritsas et al., 2023). Additional choice of a non-trivial weight function $\omega$ adds further power and flexibility. In particular, we show that even a simple weight function that maps nodes to the reciprocal of their degree is enough to exactly express RWSE in terms of MoSE (Proposition 4.5).

In the following, we will provide insight into the expressivity of MoSE by leveraging established connections between homomorphism counts and MPNNs. For Transformer architectures, similar frameworks of expressivity are not yet established. Recent work by Rosenbluth et al. (2024) shows that Transformer architectures do not inherently contribute to expressivity, and that positional/structural encoding is instead the key ingredient of their expressivity. We therefore study the expressivity of the encodings themselves, which in turn translates to the expressivity of models that utilize – and heavily depend on – these encodings. In particular, any typical Graph Transformer architecture with MoSE will naturally be at least as expressive as MoSE, and thus inherit all lower bounds from our analysis.

Our first two propositions establish that MoSE is, in a sense, incomparable to the WL-hierarchy. For every fixed set of graphs $\mathcal{G}$ that induces an encoding, we can provide an upper bound in terms of $k$-WL for some $k$ dependent on $\mathcal{G}$. At the same time, given any fixed choice of $k$, the expressiveness of even MoSE encodings induced by a single pattern graph cannot be confined to the expressive power of $k$-WL (Proposition 4.2).

**Proposition 4.1.** *Let $\mathcal{G}$ be a finite set of graphs and let $k$ be the maximum treewidth of a graph in $G$. Then, $\text{MoSE}_{\mathcal{G}}$ is at most as distinguishing as $k$-WL. That is, $\text{MoSE}_{\mathcal{G}} \preceq k\text{-WL}$.*[8]

Although we cannot distinguish *all* graphs distinguished by $k$-WL using $\text{MoSE}_{\mathcal{G}}$ with a finite $\mathcal{G}$, we can distinguish any particular pair of graphs distinguished by $k$-WL. In fact, we only need a single graph in $\mathcal{G}$ in order to do this.

**Proposition 4.2.** *For $k \geq 1$, let $G$ and $H$ be two non-isomorphic graphs that are equivalent under $k$-WL. Then, there exists a graph $F$ such that $\text{MoSE}_{\{F\}}$ distinguishes $G$ and $H$.*

Lanzinger & Barceló (2024) and Jin et al. (2024) studied the expressivity of MPNNs with respect to functions that map graphs to numbers. For a large class of these *graph motif parameters*, the distinguishing power of such a function relates closely to homomorphism counts from its homomorphism basis (Jin et al., 2024). Building on these results, we establish a very broad lower bound for MoSE-enhanced Transformers while also providing a practical method for selecting $\mathcal{G}$.

**Proposition 4.3.** *Let $f$ be a graph motif parameter with basis $\mathcal{G}_f \subseteq \mathcal{G}$. Then $\text{MoSE}_{\mathcal{G}}$ is at least as distinguishing as $f$. That is, if $\mathcal{F}$ is the family of all graph motif parameters, we have $\mathcal{F} \preceq \text{MoSE}$.*

---

[7]We require that $\mathcal{G}$ be a finite set, and that each graph within $\mathcal{G}$ have finite size.

[8]With respect to the definition of $\preceq$, both sides in this case represent singleton families of graph invariants.

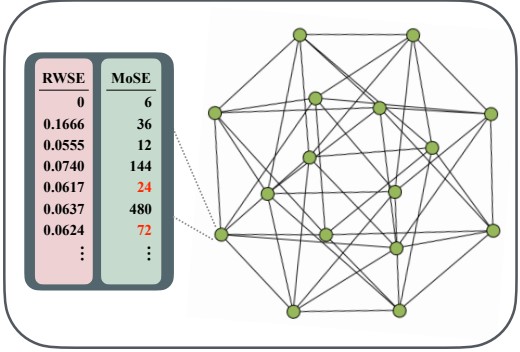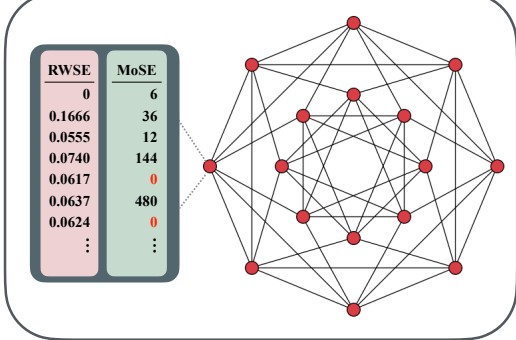

Figure 2: The $4 \times 4$ Rook's Graph (left) and the Shrikhande Graph (right) are non-isomorphic strongly regular graphs that have the same regularity parameters. RWSE produces the same vector on every vertex of both graphs whereas a simple construction of MoSE using $\mathsf{Spasm}(C_7) \cup \mathsf{Spasm}(C_8)$ homomorphism counts easily distinguishes the two graphs.

## 4.2 COMPARING THE EXPRESSIVITY OF MoSE TO RWSE

From above we see that MoSE is closely aligned to the $k$-WL hierarchy, and we can use this alignment to inform the choice of $\mathcal{G}$. However, it is not possible to entirely contain MoSE within the WL hierarchy. In the following, we show that this is not the case for RWSE. In fact, the expressiveness of RWSE, regardless of length parameter, is fully contained within 2-WL.

**Proposition 4.4.** *For every $\ell \geq 2$, any graph which can be distinguished by $RWSE_\ell$ can be distinguished by 2-WL. That is, $RWSE \preceq 2\text{-}WL$.*

This in itself has wide-ranging consequences. For instance, it is known that 2-WL cannot distinguish strongly regular graphs (Fuhlbrück et al., 2021), and therefore neither can RWSE. For MoSE, there exist no such limitations (Figure 2) beyond those dependent on the choice of $\mathcal{G}$ (Proposition 4.1).

While MoSE cannot fully capture 2-WL, it is in fact possible to express $RWSE_\ell$ for every $\ell \geq 2$ as a special case of MoSE. In contrast to previous results, our result here requires a node-weighting scheme that is not constant. However, even straightforward node-weighting by degrees – which comes at no cost in practice – is enough for our proof. Letting $\mathcal{G}$ consist of cycle graphs with size at most $\ell$ gives us the desired result (see Appendix A for details).

**Proposition 4.5.** *For any $\ell \in \mathbb{N}$, there exists a finite family of graphs $\mathcal{G}$ and a weighting scheme $\omega$ such that the node-level $RWSE_\ell(v, H)$ label is uniquely determined by $MoSE_{\mathcal{G},\omega}(v, H)$ for all $v$ and $H$. In fact, $RWSE_\ell$ is simply the special case of $MoSE_{\mathcal{G},\omega}$ where $\omega : v \mapsto 1/d(v)$ and $\mathcal{G} = \{C_i\}_{i=1}^{\ell}$.*

Proposition 4.5 immediately shows that $RWSE \preceq MoSE$, and the example from Figure 2 demonstrates that this inclusion is in fact strict.

**Theorem 4.6.** *Motif structural encoding is strictly more expressive than random walk structural encoding: $RWSE \prec MoSE$.*

Finally, we note that on the node-level, there are even cases where RWSE is weaker than 1-WL. Specifically, there are nodes for which 1-WL assigns different labels, but $RWSE_\ell$ assigns the same label for every $\ell \geq 1$ (Figure 1). There are also, of course, instances in the other way around.

**Proposition 4.7.** *RWSE is incomparable to 1-WL in terms of node-level expressiveness.*

## 5 EXPERIMENTS

In this section, we empirically demonstrate the efficacy of MoSE on several real-world and synthetic datasets across a range of models. Full experimental details and additional results are presented in Appendix B. Time and complexity details for computing MoSE encodings are in Appendix B.1.

Table 1: Performance of different positional (PE) and structural (SE) encoding types with the GPS model. MoSE consistently outperforms other encodings at relatively low computational cost.

| Encoding | PE/SE | ZINC-12K | PCQM4Mv2-subset | CIFAR10 |
|----------|-------|----------|-----------------|---------|
|          |       | MAE ↓ | MAE ↓ | Acc. ↑ |
| *none* | - | $0.113_{\pm 0.007}$ | $0.1355_{\pm 0.0035}$ | $71.49_{\pm 0.19}$ |
| PEG$^{LapEig}$ | PE | $0.161_{\pm 0.006}$ | $0.1209_{\pm 0.0003}$ | $72.10_{\pm 0.46}$ |
| LapPE | PE | $0.116_{\pm 0.009}$ | $0.1201_{\pm 0.0003}$ | $72.31_{\pm 0.34}$ |
| SignNet$^{MLP}$ | PE | $0.090_{\pm 0.007}$ | $0.1158_{\pm 0.0008}$ | $71.74_{\pm 0.60}$ |
| SignNet$^{DeepSets}$ | PE | $0.079_{\pm 0.006}$ | $0.1144_{\pm 0.0002}$ | $72.37_{\pm 0.34}$ |
| RWSE | SE | $0.070_{\pm 0.002}$ | $0.1159_{\pm 0.0004}$ | $71.96_{\pm 0.40}$ |
| **MoSE (*ours*)** | SE | $\mathbf{0.062}_{\pm 0.002}$ | $\mathbf{0.1133}_{\pm 0.0014}$ | $\mathbf{73.50}_{\pm 0.44}$ |

Table 2: We report mean absolute error (MAE) for graph regression on ZINC-12K (with edge features) across several models. MoSE yields substantial improvements over RWSE for every architecture.

Table 3: We achieve SOTA on ZINC-12K by replacing the random walk encoding from GRIT with MoSE. MP denotes if the model performs local message passing.

| Model | Encoding Type | | |
|-------|------|------|------|
|       | *none* | RWSE | **MoSE (*ours*)** |
| MLP-E | $0.606_{\pm 0.002}$ | $0.361_{\pm 0.010}$ | $\mathbf{0.347}_{\pm 0.003}$ |
| GIN-E | $0.243_{\pm 0.006}$ | $0.122_{\pm 0.003}$ | $\mathbf{0.118}_{\pm 0.007}$ |
| GIN-E+VN | $0.151_{\pm 0.006}$ | $0.085_{\pm 0.003}$ | $\mathbf{0.068}_{\pm 0.004}$ |
| GT-E | $0.195_{\pm 0.025}$ | $0.104_{\pm 0.025}$ | $\mathbf{0.089}_{\pm 0.018}$ |
| GPS | $0.119_{\pm 0.011}$ | $0.069_{\pm 0.001}$ | $\mathbf{0.062}_{\pm 0.002}$ |

| Model | MAE ↓ | MP |
|-------|-------|-----|
| GSN | $0.101_{\pm 0.010}$ | ✓ |
| CIN | $0.079_{\pm 0.006}$ | ✓ |
| GPS | $0.070_{\pm 0.002}$ | ✓ |
| Subgraphormer+PE | $0.063_{\pm 0.001}$ | ✓ |
| GT-E | $0.195_{\pm 0.025}$ | ✗ |
| GRIT+RRWP | $0.059_{\pm 0.002}$ | ✗ |
| **GRIT+MoSE (*ours*)** | $\mathbf{0.056}_{\pm 0.001}$ | ✗ |

## 5.1 COMPARING MoSE TO OTHER ENCODINGS

We begin by extending the positional and structural encoding analysis from GPS (Rampásek et al., 2022) to include MoSE. We evaluate GPS+MoSE on three different benchmarking datasets, including ZINC (Irwin et al., 2012; Dwivedi et al., 2023), PCQM4Mv2 (Hu et al., 2021), and CIFAR10 (Krizhevsky et al., 2009; Dwivedi et al., 2023). ZINC and PCQM4Mv2 are both molecular datasets, whereas CIFAR10 is an image classification dataset. Following the protocol from Rampásek et al. (2022), we use a subset of the PCQM4Mv2 dataset.

**Experimental Setup.** For ZINC, we construct MoSE$_\mathcal{G}$ using $\mathcal{G} = \mathsf{Spasm}(C_7) \cup \mathsf{Spasm}(C_8)$ in accordance with Jin et al. (2024). For the PCQM4Mv2-subset and CIFAR10 datasets, we take $\mathcal{G}$ to be the set of all connected graphs with at most 5 nodes. Details for all other encodings are described by Rampásek et al. (2022).

**Results.** Our results are presented in Table 1. The reported baselines for all encodings except MoSE are taken from Rampásek et al. (2022). We see that MoSE consistently outperforms other encoding types, including a number of spectral methods. When compared to RWSE, a prominent structural encoding, MoSE achieves over 10% relative improvement on ZINC. MoSE's strong performance on the CIFAR10 image classification benchmark highlights its effectiveness beyond molecular datasets. Additional results for MoSE on the MNIST image classification dataset (Deng, 2012; Dwivedi et al., 2023) and the LRGB Peptides datasets are in Appendix B.5 and Appendix B.6, respectively.

## 5.2 GRAPH REGRESSION ON ZINC

As mentioned above, ZINC is a molecular dataset that presents a graph-level regression task. Following Dwivedi et al. (2023), we use a subset of the ZINC dataset that contains 12,000 molecules, and we constrain all of our models to under 500k learnable parameters. We include the use of edge features in Table 2, referring readers to the appendix (Table 13) for results on ZINC without edge features. As before, we construct MoSE$_\mathcal{G}$ using $\mathcal{G} = \mathsf{Spasm}(C_7) \cup \mathsf{Spasm}(C_8)$ for all ZINC experiments.

Table 4: We report MAE results for GPS with various feature enhancements on the QM9 dataset. The best model is highlighted in red, the second best is blue, and the third best is olive.

| Property | R-GIN | R-GIN+FA | R-SPN | E-BasePlanE | GPS | |
| --- | --- | --- | --- | --- | --- | --- |
| | | | | | RWSE | **MoSE** *(ours)* |
| mu | $2.64_{\pm0.11}$ | $2.54_{\pm0.09}$ | $2.21_{\pm0.21}$ | $1.97_{\pm0.03}$ | $1.47_{\pm0.02}$ | $1.43_{\pm0.02}$ |
| alpha | $4.67_{\pm0.52}$ | $2.28_{\pm0.04}$ | $1.66_{\pm0.06}$ | $1.63_{\pm0.01}$ | $1.52_{\pm0.27}$ | $1.48_{\pm0.16}$ |
| HOMO | $1.42_{\pm0.01}$ | $1.26_{\pm0.02}$ | $1.20_{\pm0.08}$ | $1.15_{\pm0.01}$ | $0.91_{\pm0.01}$ | $0.91_{\pm0.01}$ |
| LUMO | $1.50_{\pm0.09}$ | $1.34_{\pm0.04}$ | $1.20_{\pm0.06}$ | $1.06_{\pm0.02}$ | $0.90_{\pm0.06}$ | $0.86_{\pm0.01}$ |
| gap | $2.27_{\pm0.09}$ | $1.96_{\pm0.04}$ | $1.77_{\pm0.06}$ | $1.73_{\pm0.02}$ | $1.47_{\pm0.02}$ | $1.45_{\pm0.02}$ |
| R2 | $15.63_{\pm1.40}$ | $12.61_{\pm0.37}$ | $10.63_{\pm1.01}$ | $10.53_{\pm0.55}$ | $6.11_{\pm0.16}$ | $6.22_{\pm0.19}$ |
| ZPVE | $12.93_{\pm1.81}$ | $5.03_{\pm0.36}$ | $2.58_{\pm0.13}$ | $2.81_{\pm0.16}$ | $2.63_{\pm0.44}$ | $2.43_{\pm0.27}$ |
| U0 | $5.88_{\pm1.01}$ | $2.21_{\pm0.12}$ | $0.89_{\pm0.05}$ | $0.95_{\pm0.04}$ | $0.83_{\pm0.17}$ | $0.85_{\pm0.08}$ |
| U | $18.71_{\pm23.36}$ | $2.32_{\pm0.18}$ | $0.93_{\pm0.03}$ | $0.94_{\pm0.04}$ | $0.83_{\pm0.15}$ | $0.75_{\pm0.03}$ |
| H | $5.62_{\pm0.81}$ | $2.26_{\pm0.19}$ | $0.92_{\pm0.03}$ | $0.92_{\pm0.04}$ | $0.86_{\pm0.15}$ | $0.83_{\pm0.09}$ |
| G | $5.38_{\pm0.75}$ | $2.04_{\pm0.24}$ | $0.83_{\pm0.05}$ | $0.88_{\pm0.04}$ | $0.83_{\pm0.12}$ | $0.80_{\pm0.14}$ |
| Cv | $3.53_{\pm0.37}$ | $1.86_{\pm0.03}$ | $1.23_{\pm0.06}$ | $1.20_{\pm0.06}$ | $1.25_{\pm0.05}$ | $1.02_{\pm0.04}$ |
| Omega | $1.05_{\pm0.11}$ | $0.80_{\pm0.04}$ | $0.52_{\pm0.02}$ | $0.45_{\pm0.01}$ | $0.39_{\pm0.02}$ | $0.38_{\pm0.01}$ |

**Comparing MoSE to RWSE across Multiple Architectures.** Because Rampásek et al. (2022) find that RWSE performs best on molecular datasets, we provide a detailed comparison of MoSE and RWSE across a range of architectures on ZINC. We select a baseline multi-layer perceptron (MLP-E), a simple self-attention Graph Transformer (GT-E), and GPS (Rampásek et al., 2022) for our models. Note that all models are adapted to account for edge features (see Appendix B.2 for details). For reference, we include MPNN results from GIN-E and its virtual-node extension GIN-E+VN (Hu et al., 2020). We follow Rampásek et al. (2022) and use random-walk length 20 for our RWSE benchmarks on ZINC.

**Results.** MoSE consistently outperforms RWSE across all models in Table 2. Even though the graphs in ZINC are relatively small and a random-walk length of 20 already traverses most of the graph, our experiments show that the structural information provided by MoSE yields superior performance. We also note that GIN-E+VN becomes competitive with leading Graph Transformer models when using MoSE. This highlights the effectiveness of MoSE in supplementing the benefits of virtual nodes, an MPNN-enhancement aimed at capturing long-range node interactions (Southern et al., 2025).

**Comparing MoSE to RRWP.** GRIT (Ma et al., 2023) is a recent Graph Transformer architecture that incorporates graph inductive biases without using message passing. GRIT utilizes relative random walk positional encodings (RRWP) at both the node representation level as well as the node-pair representation level. In our formulation, GRIT+MoSE, we remove all RRWP encodings for both the node and node-pair representations, and replace them with MoSE encodings at the node level.

**Results.** Using MoSE in conjunction with the GRIT architecture (Ma et al., 2023) yields state-of-the-art results on ZINC, as detailed in Table 3. Recall that we substituted MoSE for RRWP only at the node level, completely omitting pairwise encodings. This suggests that even without explicit node-pair encodings, the relevant structural information captured by MoSE still surpasses that of RRWP. This is especially interesting given that GRIT is intentionally designed to exploit the information provided by RRWP, insofar that Ma et al. (2023) present theoretical results ensuring that GRIT is highly expressive when equipped with RRWP encodings. Despite this, we show that GRIT achieves superior empirical performance when MoSE is used instead of RRWP.

## 5.3 GRAPH REGRESSION ON QM9

QM9 is a real-world molecular dataset that contains over 130,000 graphs (Wu et al., 2018; Brockschmidt, 2020). The node features include atom type and other descriptive features. Unlike ZINC, where there is only one regression target, QM9 presents 13 different quantum chemical properties to regress over, making it a much more robust benchmark.

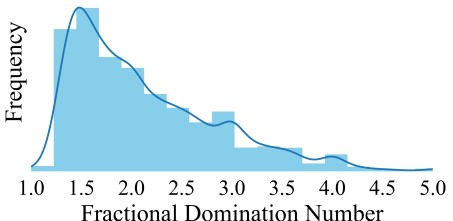

Figure 3: Plot of the distribution of fractional domination numbers in our synthetic dataset.

Table 5: MAE results for our synthetic dataset. MoSE consistently performs the best, with MLP+MoSE even outperforming the Graph Transformer with LapPE and RWSE.

| Encoding | MLP | GT |
|---|---|---|
| LapPE | $0.482_{\pm.003}$ | $0.084_{\pm.001}$ |
| RWSE | $0.075_{\pm.001}$ | $0.061_{\pm.001}$ |
| **MoSE** *(ours)* | $\mathbf{0.058}_{\pm.001}$ | $\mathbf{0.055}_{\pm.001}$ |

**Experimental Setup.** We select GPS (Rampásek et al., 2022) due to its modularity and compare GPS+RWSE to GPS+MoSE. We follow Jin et al. in constructing $\text{MoSE}_{\mathcal{G}}$ with the set of all connected graphs of at most 5 vertices and the 6-cycle: $\mathcal{G} = \Omega_{\leq 5}^{con} \cup C_6$. Since there are several regression targets for QM9, our choice of $\mathcal{G}$ is advantageous as it accounts for a diverse set of motifs, allowing GPS to express varied graph motif parameters (Proposition 4.3). We keep a random-walk length of 20 for RWSE to align with Rampásek et al. (2022). Finally, we provide several leading GNN models for comparison (Brockschmidt, 2020; Alon & Yahav, 2021; Abboud et al., 2022; Dimitrov et al., 2023).

**Results.** MoSE consistently outperforms RWSE across multiple targets, achieving the best MAE on 11 of the 13 molecular properties (Table 4). GPS+MoSE comfortably outperforms leading GNN models, including R-SPN (Abboud et al., 2022) and E-BasePlane (Dimitrov et al., 2023), the latter of which is isomorphism-complete over planar graphs. By contrast, GPS+RWSE is occasionally outperformed by these GNN architectures, despite being more computationally demanding.

### 5.4 SYNTHETIC DATASET: PREDICTING FRACTIONAL DOMINATION NUMBER

We generate a new synthetic dataset where the task is to predict each graph's *fractional domination number*. This target relies on complex long-range interactions, being an inherently global property, unlike other popular metrics (e.g., clustering coefficients) which are aggregations of local properties. The target distribution of our dataset is complex (Figure 3), making the task highly challenging. Furthermore, fractional domination is important for a range of applications, such as optimizing resource allocation (G. et al., 2024). Specific details for our dataset are given in Appendix B.9.

**Experimental Setup.** We focus on isolating the performance of MoSE compared to LapPE and RWSE. For this, we select a 4-layer MLP and a basic Graph Transformer (Dwivedi & Bresson, 2020) as our reference models. Since neither architecture contains a local message passing component, neither model contains any inherent graph inductive bias. Hence, all structure-awareness comes from the encodings. We choose $\mathcal{G} = \mathsf{Spasm}(C_7) \cup \mathsf{Spasm}(C_8)$ for $\text{MoSE}_{\mathcal{G}}$, random walk length 20 for RWSE, and dimension 8 for LapPE.

**Results.** Table 5 shows that MoSE outperforms LapPE and RWSE across both models. Strikingly, even the simple MLP+MoSE model outperforms the more advanced GT+RWSE and GT+LapPE models. These results suggest that MoSE is able to capture complex graph information in a context broader than just the molecule and image domains. We also note that MLP+LapPE performs significantly worse than all other configurations. This indicates that the self-attention mechanism of GT is a key component in allowing the model to leverage spectral information provided by LapPE.

## 6 DISCUSSION AND FUTURE WORK

We propose motif structural encoding as a flexible and powerful graph inductive bias based on counting graph homomorphisms. MoSE is supported by expressiveness guarantees that translate naturally to architectures using MoSE. In particular, we show that MoSE is more expressive than RWSE, and we relate these encoding schemes to the expressive power of MPNNs as well as the WL hierarchy more generally. We empirically validate the practical effects of our theoretical results, achieving strong benchmark performance across a variety of real-world and synthetic datasets. Although our work takes a closer look at the expressivity of different graph inductive biases, there remains several open questions (e.g., the relationship between MoSE and spectral methods). Furthermore, we have yet to explore the full potential of the node-weighting parameter in our encoding scheme.

ACKNOWLEDGMENTS

Emily Jin is partially funded by AstraZeneca and the UKRI Engineering and Physical Sciences Research Council (EPSRC) with grant code EP/S024093/1. Matthias Lanzinger acknowledges support by the Vienna Science and Technology Fund (WWTF) [10.47379/ICT2201]. This research is partially supported by EPSRC Turing AI World-Leading Research Fellowship No. EP/X040062/1 and EPSRC AI Hub on Mathematical Foundations of Intelligence: An "Erlangen Programme" for AI No. EP/Y028872/1.

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

## A  TECHNICAL DETAILS

*Proof of Proposition 4.1.* Dvořák (2010) showed that if $G$ and $H$ are equivalent under $k$-WL, then every graph $F$ with treewidth at most $k$ has $\mathsf{Hom}(F, G) = \mathsf{Hom}(F, H)$. This extends also to the vertex level. Let $v \in V(G)$ and $v' \in V(H)$ such that $v$ and $v'$ have equivalent $k$-WL labels. Then also $\mathsf{Hom}(F, G, v) = \mathsf{Hom}(F, H, v')$ (see Lanzinger & Barceló (2024), Lemma 12). Hence, if the maximum treewidth in $\mathcal{G}$ is $k$, then equivalence under $k$-WL implies also that they are equivalent under $\mathsf{MoSE}_{\mathcal{G}}$. $\qquad\square$

*Proof of Proposition 4.2.* It is well known (Lovász, 1967) that for every two non-isomorphic graphs $G$ and $H$, there is a graph $F$ such that $\mathsf{Hom}(F, G) \neq \mathsf{Hom}(F, H)$. We have that $\mathsf{MoSE}_{\{F\}}$ will distinguish the two graphs. $\qquad\square$

*Proof of Proposition 4.3.* Suppose the opposite, i.e., there is a pair of graphs $G, H$ such that $f(G) \neq f(H)$, but $\mathsf{MoSE}_{\Gamma_f}$, where $\Gamma_f$ is set of graphs in the basis of $f$, creates the same multiset of labels on both graphs. Then specifically, also $\mathsf{Hom}(F, G) = \mathsf{Hom}(F, H)$ for every $F \in \Gamma_f$. But since $f$ is a graph motif parameter, $f(G) = f(H)$ and we arrive at a contradiction. $\qquad\square$

### A.1  THEOREM 4.6

*Proof of Proposition 4.5.* Take any graph $H$ and notate $V(H) = \{1, 2, ..., n\}$. For $v, v' \in V(H)$, define $P(v \xrightarrow{i} v')$ to be the probability of starting a length $i$ random walk at node $v$ and ending it at node $v'$. Here, "length $i$" refers to the number of edges in the random walk where we allow repeat edges, and "random walk" refers to a walk where each step (say we are currently at node $v$) assigns a uniform probability $1/d(v)$ to moving to any of the neighbors in $\mathcal{N}(v)$.

As shorthand, define $M = D^{-1}A$ where $D$ and $A$ are the diagonal degree matrix and adjacency matrix of $H$ respectively. Let us first prove that $(M^i)_{v,v'} = P(v \xrightarrow{i} v')$ for any nodes $v, v' \in V$ and for all powers $i \in \mathbb{N}$ by performing induction on $i$. The base case $i = 1$ holds trivially because we assume that random steps are taken uniformly over neighbors. Assume as the inductive hypothesis that our claim holds for some $i \geq 1$, and note that:

$$(M^{i+1})_{v,v'} = (M^1 M^i)_{v,v'} = \sum_{x=1}^{n} M_{v,x} \cdot M^i_{x,v'}$$

but our base case and inductive hypothesis tell us that

$$\sum_{x=1}^{n} M_{v,x} \cdot M^i_{x,v'} = \sum_{x=1}^{n} P(v \xrightarrow{1} x) \cdot P(x \xrightarrow{i} v')$$

By the law of total probability:

$$\sum_{x=1}^{n} P(v \xrightarrow{1} x) \cdot P(x \xrightarrow{i} v') = P(v \xrightarrow{i+1} v')$$

This completes the induction. Hence, for any node $v$ in any graph $H$, the node-level $\text{RWSE}_k$ vector is equivalent to:

$$\text{RWSE}_k(v, H) = \left[ (M^i)_{v,v} \right]_{i=1}^{k} = \left[ P(v \xrightarrow{i} v) \right]_{i=1}^{k} \in \mathbb{R}^k$$

Now, let us show that we can recover this random-walk vector using MoSE. Consider the family of all cycles with up to $k$ nodes $\mathcal{G} = \{C_i\}_{i=1}^{k}$, and let $\omega$ be the vertex weighting which maps any node to the reciprocal of its degree. Of course, the "cycles" $C_1$ and $C_2$ are not well-defined, so let us set $C_1$ to be the empty graph (with no nodes and no edges) and $C_2$ to be the graph with two nodes connected by an edge. We say that there are zero homomorphisms from $C_1$ into any other graph purely as a notational convenience aligning with the fact that $(M^1)_{v,v}$ is always 0.

For each $C_i \in \mathcal{G}$ with $i > 1$, enumerate the nodes $V(C_i) = \{u_0, u_1, ..., u_{i-1}\}$ such that $(u_\ell, u_{\ell+1}) \in E(C_i)$ for all $\ell = 0, ..., i-2$ and $(u_{i-1}, u_0) \in E(C_i)$. Take any node $v$ in any graph $H$, and define $\mathcal{H}_i$ to be the set of all homomorphisms from $C_i$ to $G$ which map node $u_0 \in C_i$ to node $v \in G$. Each homomorphism in $f \in \mathcal{H}_i$ can be constructed by making $i - 1$ choices of node image $f(u_\ell) \in \mathcal{N}(f(u_{\ell-1}))$ for $\ell = 1, ..., i-1$ such that $f(u_{(i-1)}) \in \mathcal{N}(f(u_0))$ where we have fixed $f(u_0) = v$. In other words, each homomorphism $f \in \mathcal{H}_i$ corresponds to a $i$-edge walk (allowing edge repeats) starting and ending at node $v$, which we will call $W_f$. This correspondence is described by taking the induced subgraph $W_f = G[f(C_i)]$ where $f(C_i)$ is the image of $C_i$ under homomorphism $f$. In the reverse direction, any $i$-edge walk

$$W_f: \quad v = v^{(0)}, v^{(1)}, ...., v^{(i-1)}, v^{(i)} = v \quad \text{in } H$$

that starts and ends at node $v$ corresponds to a homomorphism $f \in \mathcal{H}_i$ which maps $f(u_\ell) = v^{(\ell)}$. Hence, we have described a bijective correspondence between $\mathcal{H}_i$ and the set of $i$-edge walks starting/ending at $v$.

For any $f \in \mathcal{H}_i$, we have:

$$\prod_{v \in V(C_i)} \omega(f(v)) = \prod_{v \in W_f} 1/d(v) = \prod_{\ell=0}^{i-1} 1/d(v^{(\ell)})$$

Since the probability of stepping from node $v^{(\ell)} \in V(W_f)$ to node $v^{(\ell+1)} \in V(W_f)$ in a random walk is simply $1/d(v^{(\ell)})$, it holds that:

$$\prod_{\ell=0}^{i-1} 1/d(v^{(\ell)}) = P(W_f)$$

where $P(W_f)$ is the probability of performing the random walk $W_f$. Thus,

$$\omega\text{-Hom}_{u_0 \to v}(C_i, G) = \sum_{f \in \mathcal{H}_i} P(W_f) = P(v \xrightarrow{i} v)$$

giving us:

$$\text{MoSE}_{\mathcal{G},\omega}(v, H) = \left[\omega\text{-Hom}_{\to v}(C_i, H)\right]_{i=1}^d = \left[P(v \xrightarrow{i} v)\right]_{i=1}^k = \text{RWSE}_k(v, H)$$

$\square$

*Proof of Proposition 4.4.* We first recall the walk refinement procedure from Lichter et al. (2019) such that we can relate it directly to RWSE.

The scheme at its basis assumes a directed complete colored graph $G = (V, H, \chi)$ where the coloring $\chi$ always assigns different colors to self-loops than to other edges. For tuples of $m$ vertices $(v_1, v_2, \ldots, v_m) \in V^m$, we define

$$\bar{\chi}(v_1, v_2, \ldots, v_m) = ((\chi(v_1, v_2), \chi(v_2, v_3), \ldots, \chi(v_{m-1}, v_m)).$$

Ultimately, we want to use the scheme on undirected uncolored graphs. An undirected (and uncolored) graph $G$, is converted into the setting above by adding the coloring $\chi : V(G) \to \{-1, 0, 1\}$ as follows: for all self-loops $\chi$ assigns $-1$, i.e., $\chi(v, v) = -1$. For all distinct pairs $v, u$, we set $\chi(v, u) = 1$ if $(v, u) \in E(G)$ and $\chi(v, u) = 0$ if $(v, u) \notin E(G)$. For the directed edges recall that we consider complete directed graphs and hence $E = V^2$.

With this in hand we can define the walk refinement procedure of Lichter et al. (2019). Formally, for $k \geq 2$, the $k$-walk refinement of a colored complete directed graph $G = (V, E, \chi)$ as the function that gives the new coloring $\chi_{W[k]}$ to every edge is as follows

$$\chi_{W[k]}(v, u) = \{\!\!\{\bar{\chi}(v, w_1, \ldots, w_{k-1}, u) \mid w_i \in V\}\!\!\}$$

Intuitively, the refinement takes into account all walks of length $k$ in the graph. Note that walks of length shorter than $k$ are also captured via self-loops (because of their different weights we are also always aware that a self-loop is taken and that a specific tuple in the multi-set represents a shorter walk. As with the Weisfeiler-Leman color refinement procedure, $k$-walk refinement can be iterated until it reaches a stable refinement (after finitely many steps). We will write $\chi^i(v, u)$ to designate the color obtained after $i$ steps of the refinement (with $\chi(v, u) = \chi^0(v, u)$). Importantly, Lichter et al. (2019) (Lemma 4) show that the stable refinements of the $k$-walk refinement procedure is always reached after finitely many steps and produces the same partitioning of vertices as 2-WL. That is, any pairs $(u, v)$ and $(x, y)$ that receive the same label by 2-WL refinement, also have $\chi^\infty(v, u) = \chi^\infty(x, y)$.

All that is then left is to show that the stable $W[k]$ refinement uniquely determines the $\text{RWSE}_k$ feature vector for every vertex $v$. In particular, we show that every entry of the $\text{RWSE}_k$ vector of any vertex $v$ can be uniquely derived from the $W[k]$ labeling of the pair $(v, v)$. As shown in Proposition 4.5, the $k'$-th entry of the $\text{RWSE}_k$ vector for $v$, is uniquely determined by the number of $k'$-hop paths beginning and ending in $v$, together with the degrees along every such path. In the following, we observe that this combination of information can be directly computed from the $W[k]$ labeling of $(v, v)$.

We first observe that $\chi^1_{W[k]}(v, v)$ uniquely determines the degree of $v$. It is equivalent to the number of $k$-walks that move to an adjacent vertex and continue with self-loops for the $k - 1$ remaining steps. That is:

$$\text{degree}(v) = \text{degree}(\chi^1_{W[k]}(v, v)) := |\{(1, -1, -1, \ldots, -1) \in \chi^1_{W[k]}(v, v)\}| \tag{2}$$

Similarly, it is straightforward to recognise the original label $\chi(v, u)$ from $\chi^1_{W[k]}(v, u)$:

$$\chi(v, u) = -1 \iff \{-1\}^k \in \chi^1_{W[k]}(v, u) \tag{3}$$

that is $v = u$ if and only if $u$ can be reached from $v$ only through self-loops. Similarly we can determine the other labels, and we only state the case for an edge existing here:

$$\chi(v, u) = 1 \iff (1, -1, -1, \ldots, -1) \in \chi^1_{W[k]}(v, u). \tag{4}$$

We can similarly retrieve the label $\chi(v, u)$ from $\chi^2_{W[k]}(v, u)$, since in any tuple of the form

$$((\chi(v_1, v_2), \chi(v_2, v_3), \ldots, \chi(v_{m-1}, v_m))$$

we know by Equation (3) and Equation (4) whether the respective pair of vertices are adjacent or equal. We can then naturally iterate the definitions from Equation (3) and Equation (4). Thus, for every $1 \leq k' \leq k$, we can directly enumerate all paths from $v$ to $v$ by inspecting the multiset $\chi^2(v, v)$.

To complete the argument outlined above, we need to show that for every $v, u$, $\chi^2(v, u)$ determines the degree of $v$. Now as above we can determine $\chi(v, u)$ and for each case observe how to obtain the degree. We have $\chi(v, u) = -1$, if and only if there is a tuple for the walk $(v, v, \ldots, v) \in \chi^2(v, u)$ (and the tuple can be recognised). The degree is then directly determined by the first tuple through Equation (2). Similarly, $\chi(v, u) = 1$, if and only if there is a tuple $(v, u, \ldots, u) \in \chi^2(v, u)$ that corresponds to original labels $(-1, -1, \ldots, -1, 1)$. We can thus again obtain the degree from the first position of this tuple, and analogously for the 0 label.

In summary, we see that after 2 steps of $k$-walk refinement, every label of $(v, v)$ has enough information to uniquely determine all 1 to $k$-walks from $v$ to $v$, together with the degree of every node of the walk. As seen in the proof of Proposition 4.5, this determines the $\mathrm{RWSE}_k$ vector. $\qquad\square$

We note that the proof in fact shows a stronger statement than $\mathrm{RWSE}_k$ being at most as expressive as 2-WL. It shows that $\mathrm{RWSE}_k$ is at most as expressive as 2-steps of $k$-walk refinement. This can be further related to equivalence under a certain number of 2-WL steps through the work of Lichter et al. (2019).

*Proof of Proposition 4.7.* One direction is shown in Figure 1, which shows two graphs $G$ and $H$, with highlighted green and black nodes each. For every step length $\ell$, RWSE produces the same node features for both green vertices $(u_1, u_2)$ and both black vertices $(v_1, v_2)$, respectively. At the same time, it is straightforward to see that the 1-WL labeling of the green/black nodes is different in the two graphs.

For the other direction, we use the classic example comparing the graph $G$ consisting of two triangles and the graph $H$ containing the 6-vertex cycle as drawn below (Figure 4). It is well known that all nodes in both graphs have the same 1-WL label. In contrast, $\mathrm{RWSE}_3$ labels the nodes in $G$ differently from the nodes in $H$, and the respective vectors are given below. $\qquad\square$

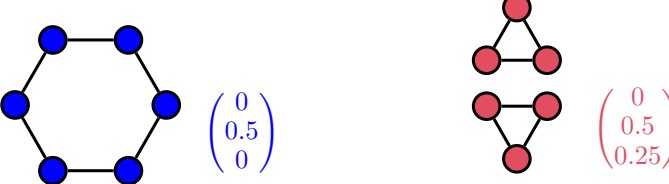

Figure 4: The 6 vertex cycle (left) and the disjoint sum of 2 triangles (right) are indistinguishable by 1-WL, but distinguishable by RWSE already after 3 steps. The RWSE vectors in the respective graphs are all the same and are given next to the graphs.

# B  ADDITIONAL EXPERIMENTAL DETAILS

We provide additional experimental details and hyperparameters for our results in Section 5. All code and instructions on how to reproduce our results are available at the following link: https://github.com/linusbao/MoSE.

**Compute Resources.**   All experiments were conducted on a cluster with 12 NVIDIA A10 GPUs (24 GB) and 4 NVIDIA H100s. Each node had 64 cores of Intel(R) Xeon(R) Gold 6326 CPU at 2.90GHz and  500GB of RAM. All experiments used at most 1 GPU at a time.

## B.1  COMPLEXITY OF COMPUTING HOMOMORPHISM COUNTS FOR MoSE

Practically, computing the homomorphism counts for MoSE is highly parallelizable since the homomorphisms from each pattern can be counted independently from one another. In our experiments,

Table 6: Time to compute MoSE embeddings used in all experiments reported as elapsed real time (wall time). Reported times are normalized by dataset size (i.e. per graph). $\Omega_{\leq 5}^{con}$ refers to the set of all connected graphs with at most 5 nodes, and $C_i$ is the cycle graph on $i$ nodes.

| Dataset | # Graphs | Avg. # nodes | Avg. # edges | Motifs | Time (sec) |
|---|---|---|---|---|---|
| ZINC | 12,000 | 23.2 | 49.8 | $\mathsf{Spasm}(C_7) \cup \mathsf{Spasm}(C_8)$ | 0.03 |
| QM9 | 130,831 | 18.0 | 37.3 | $\Omega_{\leq 5}^{con} \cup C_6$ | 0.02 |
| PCQM4Mv2-subset | 446,405 | 14.1 | 14.6 | $\Omega_{\leq 5}^{\overline{con}} \cup C_6$ | 0.03 |
| CIFAR10 | 60,000 | 117.6 | 941.1 | $\Omega_{\leq 5}^{con}$ | 0.08 |
| MNIST | 70,000 | 70.6 | 564.5 | $\Omega_{\leq 5}^{\overline{con}}$ | 0.06 |
| Synthetic | 10,000 | 23.5 | 137.9 | $\mathsf{Spasm}(C_7) \cup \mathsf{Spasm}(C_8)$ | 0.06 |
| Peptides-func/struct | 15,535 | 150.9 | 307.3 | Both* | 0.01 |

*The "Both" motif refers to $\mathsf{Spasm}(C_7) \cup \mathsf{Spasm}(C_8)$ and $\Omega_{\leq 5}^{con} \cup C_6$, benchmarked seperately.

computing the MoSE encodings took only a fraction of a second per graph (see Table 6). By contrast, training various Transformer models on these datasets took between 15-30 hours for a single run. Hence, the time to compute MoSE is almost negligible overall.

From a theoretical perspective, the complexity of MoSE is very intricate. With constant uniform weighting, counting homomorphisms from $H$ to $G$ is feasible in time $O(|G|^{tw(H)})$ where $tw(H)$ is the treewidth of $H$. It is known that this exponent cannot be substantially improved upon (Marx, 2010). The complexity of computing MoSE encodings thus fundamentally depends on the choice of patterns and their treewidth.

As we show in Proposition 4.5, RWSE is simply a limited special case of MoSE for one specific set of pattern graphs (cycles) and one specific weighting scheme ($\omega(v) = 1/d(v)$). Cycles always have treewidth 2, and thus RWSE is always feasible in quadratic time. However, RWSE cannot capture other treewidth 2 motifs that MoSE can accommodate for essentially the same cost.

## B.2 MODEL DEFINITIONS

We give definitions for those architectures which are not taken directly from the literature. Given a graph $G$ and a node $v \in V(G)$, let the node representation $\boldsymbol{h}_v^{(0)} \in \mathbb{R}^d$ be a vector concatenation of the initial node label and the structural/positional encoding of $v$. To apply an $L$-layer MLP on $G$, we update each node representation in parallel for $t = 1, ..., L$:

$$\boldsymbol{h}_v^{(t)} = \mathrm{ReLU}\left(\boldsymbol{W}^{(t)}\boldsymbol{h}_v^{(t-1)} + \boldsymbol{b}^{(t)}\right), \qquad \forall\, v \in V(G) \tag{5}$$

When edge features are available, MLP-E applies two MLPs in parallel to process node representations and edge representations separately. That is, node representations $\boldsymbol{h}_v^{(t)}$ are learned according to equation (5) using a set of weights $\{\boldsymbol{W}_{\mathrm{node}}^{(t)}, \boldsymbol{b}_{\mathrm{node}}^{(t)}\}_{t=1}^L$, while the edge representations $\boldsymbol{e}_{u,v}^{(t)} \in \mathbb{R}^{d_e}$ are learned using a different set of weights:

$$\boldsymbol{e}_{u,v}^{(t)} = \mathrm{ReLU}\left(\boldsymbol{W}_{\mathrm{edge}}^{(t)}\boldsymbol{e}_{u,v}^{(t-1)} + \boldsymbol{b}_{\mathrm{edge}}^{(t)}\right), \qquad \forall\, (u,v) \in E(G)$$

for $t = 1, ..., L_e$. Note that we *do not* require that $d = d_e$ and $L = L_e$.

The GT model updates node representations by using scaled dot-product attention (Vaswani et al., 2017):

$$\boldsymbol{h}_v^{(t)} = \mathrm{MLP}^{(t)}\left(\boldsymbol{W}_O^{(t)} \bigoplus_{h=1}^{n_H} \boldsymbol{W}_{V,h}^{(t)} \sum_{u \in V(G)} \alpha_{v,u,h}^{(t)} \boldsymbol{h}_u^{(t-1)}\right) \tag{6}$$

$$\alpha_{v,u,h}^{(t)} = \mathrm{softmax}_u\left(\frac{\boldsymbol{W}_{Q,h}^{(t)}\boldsymbol{h}_v^{(t-1)} \cdot \boldsymbol{W}_{K,h}^{(t)}\boldsymbol{h}_u^{(t-1)}}{\sqrt{d}}\right) \tag{7}$$

for $t = 1, ..., L$. Here, the $\oplus$ symbol denotes vector concatenation, and softmax is computed relative to the vector formed by iterating the expression inside the softmax argument over $u \in V(G)$. When we are provided with edge labels that come from a finite collection of discrete edge types, the GT-E models uses a learnable look-up embedding to learn a representation $\boldsymbol{e}_{u,v,h}^{(t)} \in \mathbb{R}^d$ for every node pair $u \neq v \in V(G)$, every attention head $h = 1, ..., n_H$, and every layer $t = 1, ..., L$. Non-adjacent node pairs $(u, v) \notin E(G)$ are assigned some pre-defined "null edge-type". Then, these edge representations bias the attention as:

$$\alpha_{v,u,h}^{(t)} = \text{softmax}_u \left( \frac{\text{dot}\left[\boldsymbol{W}_{Q,h}^{(t)} \boldsymbol{h}_v^{(t-1)}, \boldsymbol{W}_{K,h}^{(t)} \boldsymbol{h}_u^{(t-1)}, \boldsymbol{e}_{u,v,h}^{(t)}\right]}{\sqrt{d}} \right) \tag{8}$$

where

$$\text{dot}[\boldsymbol{a}, \boldsymbol{b}, \boldsymbol{c}] = \sum_i \boldsymbol{a}_i \cdot \boldsymbol{b}_i \cdot \boldsymbol{c}_i$$

The node update equation of GT-E is defined by simply plugging in the edge-biased attention values $\alpha_{v,u,h}^{(t)}$ given by equation (8) into the node update equation (6). Our method of "biasing" attention values with edge representations (eq. 8) is inspired by the edge-type-aware graph transformer models of Dwivedi & Bresson (2020) and Kreuzer et al. (2021).

The final node representations are sum pooled in order to produce a graph representation $\boldsymbol{h}_G$. The MLP-E model additionally pools edge representations and concatenates the edge pool to the node pool:

$$\boldsymbol{h}_G = \sum_{v \in V(G)} \boldsymbol{h}_v^{(L)} \qquad \text{for MLP, GT, and GT-E}$$

$$\boldsymbol{h}_G = \left( \sum_{v \in V(G)} \boldsymbol{h}_v^{(L)} \right) \oplus \left( \sum_{(u,v) \in E(G)} \boldsymbol{e}_{u,v}^{(L_e)} \right) \qquad \text{for MLP-E}$$

Finally, an MLP prediction head is applied on $\boldsymbol{h}_G$ to produce an output label. The reader should note that equations (5)-(8) only describe the essential aspects of our models; when we implement these models in practice, we additionally include the usual regularization/auxiliary modules at each layer (e.g., batch normalization and dropout), as specified by the hyperparameter details below.

## B.3 HYPERPARAMETER PROTOCOL

For all experiments which compare multiple positional/structural encodings, we perform the same grid-search on each encoding independently (however, this does not necessarily apply to reference results that we draw from other works). For some of our experiments, we take the shortcut of grid searching a wider range of values on the baseline *none*-encoding (where we use the model *without* any positional/structural encoding), and then we proceed to search only a strict subset of (high-performing) values for each non-empty encoding (e.g., MoSE, RWSE, and LapPE). In this case, since the grid search for RWSE/MoSE/LapPE is strictly less exploratory than our baseline search, our reported results for RWSE/MoSE/LapPE are (at worst) a conservative *under-estimate* of the improvement they provide over the *none*-encoding baseline, or (at best) a perfectly fair comparison.

See the appendix sections that follow for details on our hyperparameter configurations and hyperparameter searches. We try to give as many relevant specifics as possible here in the appendix, but some of the hyperparameter details (especially those for our large Transformer models) are excessive for the purposes of this appendix, and are much better expressed in a model configuration file. Hence, we only outline the most important details here, asking readers to reference the online code repository (https://github.com/linusbao/MoSE) for all other information.

**Blanket Hyperparameters.** There are number of hyperparameters which remain fixed across *all* of our experiments, so we state them here once to avoid redundancy. In all models, a 2-layer MLP is used to embed the raw positional/structural encoding values. The dimension of this embedding scales according to the width of the main model being used (see the code repository for exact values). Unless otherwise specified, all models utilize global sum-pooling followed by a 3-layer MLP prediction head where each layer subsequently reduces the hidden dimension by a factor of 1/2 (other than the output layer, which has a dimension equal to the number of targets). Lastly, all models use the ReLU activation function.

## B.4 EXPERIMENTAL DETAILS FOR ENCODING COMPARISON ANALYSIS

We give experimental details for the MoSE results shown in Table 1. All of our MoSE results in Table 1 are the mean (± standard deviation) of 4 runs with different random seeds. Table 1 results for encodings other than MoSE are taken from Rampásek et al. (2022).

### B.4.1 ZINC-12K

We use a subset of the ZINC molecular dataset that contains 12,000 graphs (Dwivedi et al., 2023). The dataset is split into 10,000 graphs for training, 1,000 graphs for validation, and 1,000 graphs for testing. Each graph in the dataset represents a molecule, where the node features indicate the atom type, and the edge features indicate the bond type between two atoms. The dataset presents a graph-level regression task to predict the constrained solubility of a given molecule. Following Rampásek et al. (2022), we constrain all models to a 500K parameter budget.

**GPS**   Hyperparameter details for GPS+MoSE results on ZINC in Table 1 are identical to those used in Appendix B.7.3.

### B.4.2 PCQM4Mv2-SUBSET

PCQM4Mv2-subset is a subset of the Large-Scale Open Graph Benchmark PCQM4Mv2 dataset (Hu et al., 2021) . As mentioned in Rampásek et al. (2022), this subset samples the original PCQM4Mv2 dataset as follows: 10% of the original training set, 33% of the original validation set, the entire test set. This results in 322,869 training graphs, 50,000 validation graphs, and 73,545 test graphs, for a total dataset size of 446,405 molecular graphs. In order to align with Rampásek et al. (2022), use the exact same PCQM4Mv2-subset graphs that they use in their original ablation study.

Similarly to ZINC, PCQM4Mv2 is a molecular dataset, where each graph represents a molecule, with nodes denoting atoms and edges denoting bonds. The dataset presents a graph-level regression task to predict the DFT-calculated HOMO-LUMO energy gap of a given molecule.

**GPS**   We do not perform a structured grid search for GPS+MoSE results on PCQM4Mv2 in Table 1. Instead, we simply try a few different model configurations (which can be found in our code repository) inspired by our experiments on ZINC and QM9. The final model used in our reported results is summarized as follows:

| | | |
|---|---|---|
| *MPNN Type*: GatedGCN | *Attention Type*: Standard* | *Layers*: 12 |
| *Width/Heads*: 256 / 4 | *Batch Size*: 256 | *Optimizer*: AdamW |
| *Weight Decay*: 0 | *Base LR*: 0.0002 | *Max Epochs*: 300 |
| *LR Scheduler*: Cosine | *LR Warmup Epochs*: 10 | *Attention Dropout*: 0.1 |

*The "Standard" attention type refers to multi-head scaled dot product attention.

### B.4.3 CIFAR10

The original CIFAR10 dataset consists of 60,000 32x32 color images in 10 classes, with 6,000 images in each class (Krizhevsky et al., 2009). From this, Dwivedi et al. (2023) generate a graph benchmarking dataset by constructing an 8 nearest-neighbor graph of SLIC superpixels for each image. The graph benchmarking dataset maintains the same splits as the original image dataset, which contains 45,000 training, 5,000 validation, and 10,000 test graphs. Following Rampásek et al. (2022), we constrain our reported model to ∼100K tunable parameters.

**GPS**   We grid search GPS+MoSE on CIFAR10 (Table 1) as follows. First, we fix the following hyperparameters:

| | | |
|---|---|---|
| *MPNN Type*: GatedGCN | *Attention Type*: Standard | *Attention Heads*: 4 |
| *Batch Size*: 16 | *Base LR*: 0.001 | *Max Epochs (warmup)*: 100 (5) |
| *LR Scheduler*: Cosine | *Optimizer*: AdamW | *Attention Dropout*: 0.5 |

Then, we grid-search all combinations of the following: (the best configuration with $\sim$100K parameters is shown in **bold**)

Layers: $\{2, \mathbf{3}\}$     Width: $\{36, \mathbf{52}, 72\}$     Weight Decay: $\{10^{-3}, \mathbf{10^{-4}}, 10^{-5}\}$

Dropout of Feed-forward and MPNN Layers: $\{0.0, \mathbf{0.2}, 0.4\}$

**Additional Results**  Our reported results in Table 1 use homomorphism counts from the family $\Omega^{con}_{\leq 5}$. In Table 7 below, we also report results for MoSE constructed from $\mathsf{Spasm}(C_7) \cup \mathsf{Spasm}(C_8)$, as well as results that exceed the 100K parameter budget (hyperparameter details for these extra experiments can be found in the code repository).

Table 7: We benchmark GPS+MoSE on CIFAR10. Our pattern families are $\mathcal{S} = \mathsf{Spasm}(C_7) \cup \mathsf{Spasm}(C_8)$ and $\Omega = \Omega^{con}_{\leq 5}$. GPS+LapPE results from Rampásek et al. (2022) are included for reference. MoSE outperforms LapPE across the board, and larger models perform best.

| Model | CIFAR10 (Acc. $\uparrow$) | Parameters |
|---|---|---|
| GPS+LapPE | $72.31_{\pm 0.34}$ | 112,726 |
| GPS$_{\text{small}}$+MoSE$_{\mathcal{S}}$ *(ours)* | $73.34_{\pm 0.50}$ | 110,188 |
| **GPS$_{\text{big}}$+MoSE$_{\mathcal{S}}$** *(ours)* | $\mathbf{75.00}_{\pm 0.59}$ | 267,134 |
| GPS$_{\text{small}}$+MoSE$_{\Omega}$ *(ours)* | $73.50_{\pm 0.44}$ | 114,330 |
| GPS$_{\text{big}}$+MoSE$_{\Omega}$ *(ours)* | $74.73_{\pm 0.64}$ | 215,794 |

## B.5   MNIST

Due to the success of MoSE on CIFAR10, we conduct an additional experiment with GPS+MoSE on the MNIST image classification dataset. The original MNIST dataset consists of 70,000 28x28 black and white images in 10 classes, representing handwritted digits (Deng, 2012). From this, Dwivedi et al. (2023) follow the same procedure as for CIFAR10 and generate a graph benchmarking dataset by constructing an 8 nearest-neighbor graph of SLIC superpixels for each image. The graph dataset also has the same original dataset splits of 55,000 training, 5,000 validation, and 10,000 test graphs.

In Table 8, we compare GPS+MoSE$_{\Omega^{con}_{\leq 5}}$ with other leading GNN and Graph Transformer models, and we follow Rampásek et al. (2022) in constraining our model to $\sim$100K tunable parameters. In Table 9, we report additional GPS+MoSE results using counts from $\mathsf{Spasm}(C_7) \cup \mathsf{Spasm}(C_8)$, as well as GPS+MoSE models that exceed the 100K parameter budget. We do not grid-search GPS+MoSE on MNIST, and instead we simply extrapolate from our grid-search on CIFAR10 to infer our model configurations (see code repository for details). All results for GPS+MoSE are the mean ($\pm$ standard deviation) of 4 runs with different random seeds, and all results for other models are taken from Rampásek et al. (2022). We observe that GPS+MoSE is competitive with leading GNNs on MNIST.

Table 8: Results on MNIST with $\sim$100K parameter models. Best results in red, second best in blue.

| Model | Acc. $\uparrow$ |
|---|---|
| GIN | $96.485_{\pm 0.252}$ |
| GAT | $95.535_{\pm 0.205}$ |
| CRaW1 | $97.944_{\pm 0.050}$ |
| GRIT | $98.108_{\pm 0.111}$ |
| **EGT** | $98.173_{\pm 0.087}$ |
| GPS(+LapPE) | $98.051_{\pm 0.126}$ |
| ***GPS+MoSE** | $98.135_{\pm 0.002}$ |

\* denotes *our* model.

Table 9: GPS+MoSE on MNIST with various configurations. We include EGT and GPS+LapPE for reference. $\mathcal{S} = \mathsf{Spasm}(C_7) \cup \mathsf{Spasm}(C_8)$ and $\Omega = \Omega^{con}_{\leq 5}$. Best results in red, second best in blue.

| Model | Acc. $\uparrow$ | $\sim$Parameters |
|---|---|---|
| EGT | $98.173_{\pm 0.087}$ | 100K |
| GPS+(LapPE) | $98.051_{\pm 0.126}$ | 100K |
| *GPS$_{\text{small}}$+MoSE$_{\mathcal{S}}$ | $98.090_{\pm 0.050}$ | 100K |
| ***GPS$_{\text{big}}$+MoSE$_{\mathcal{S}}$** | $98.273_{\pm 0.075}$ | 400K |
| *GPS$_{\text{small}}$+MoSE$_{\Omega}$ | $98.135_{\pm 0.002}$ | 100K |
| ***GPS$_{\text{big}}$+MoSE$_{\Omega}$** | $98.198_{\pm 0.180}$ | 200K |

\* denotes *our* model.

## B.6 Results on Peptides-struct and Peptides-func Datasets

The Peptides-func and Peptides-struct datasets were introduced in the Long Range Graph Benchmark (Dwivedi et al., 2022b). Similar to other chemistry benchmarks, each graph represent a peptide, where nodes denote atoms and edges denote the bonds between atoms. However, peptides are generally much larger than the small drug-like molecules in other datasets, such as ZINC.

Peptides-func presents a 10-class multilabel classification task, whereas Peptides-struct presents an 11-target regression task. We compare the performance of MoSE with motif patterns $\mathsf{Spasm}(C_7) \cup \mathsf{Spasm}(C_8)$, MoSE with motifs $\Omega_{\leq 5}^{con} \cup C_6$, and RWSE of length 8 because each of these three encoding have similar "reach" (i.e., the edge-radius of each node encoding's receptive field). In order to isolate the power of our encodings, we decide to use a standard self-attention Graph Transfer Dwivedi & Bresson (2020) as our reference model. Since this model does not contain any built-in graph inductive biases, all expressive power is derived from the encodings. Also, note that GT does *not* utilize the edge features provided in the Peptides dataset because GT does not receive the adjacency matrix as an explicit input. Table 10 shows that MoSE once again outperforms RWSE on both Peptides-func and Peptides-struct.

Table 10: We benchmark on the peptides-struct and peptides-func datasets using self-attention Graph Transformer to isolate the performance gains of MoSE vs RWSE. Our pattern families are $\mathcal{S} = \mathsf{Spasm}(C_7) \cup \mathsf{Spasm}(C_8)$ and $\Omega = \Omega_{\leq 5}^{con} \cup C_6$. Reported results are the mean ($\pm$ std dev.) of 4 runs with different random seeds. Best results are in red, second best in blue.

| Model | Peptides-struct (MAE $\downarrow$) | Peptides-func (AP $\uparrow$) |
|---|---|---|
| GT+none | $0.454_{\pm 0.027}$ | $0.447_{\pm 0.017}$ |
| GT+RWSE | $0.344_{\pm 0.009}$ | $0.625_{\pm 0.012}$ |
| **GT+MoSE$_{\mathcal{S}}$** *(ours)* | $0.339_{\pm 0.007}$ | $0.635_{\pm 0.011}$ |
| **GT+MoSE$_{\Omega}$** *(ours)* | $0.318_{\pm 0.010}$ | $0.629_{\pm 0.011}$ |

**GT** We do not perform hyperparameter tuning on Peptides-func, using a 4-layer GT with width 96 for all encoding types. On Peptides-struct, the training behavior is slightly more erratic, so we conduct a minimal hyperparameter search for each encoding (we search the same configurations on each encoding independently for fair comparison). We omit the details here, but the reader may reference the code repository for exact configurations. Table 11 gives a summary of our hyperparameters on Peptides-struct. All models (for both tasks) are trained for 200 epochs using the AdamW optimizer, a batch size of 128, and a base LR of 0.0003 (annealed with a cosine scheduler).

Table 11: A summary of the GT hyperparameters used for Peptides-struct results.

| Model | Layers | Width | Encoding dim |
|---|---|---|---|
| GT+none | 4 | 96 | N/A |
| GT+RWSE | 1 | 36 | 16 |
| GT+MoSE$_{\mathcal{S}}$ *(ours)* | 1 | 36 | 24 |
| GT+MoSE$_{\Omega}$ *(ours)* | 1 | 36 | 16 |

## B.7 ZINC Experimental Details & Additional Results

We use the same ZINC dataset setup as described above in Appendix B.4.1. All reported results are the mean ($\pm$ standard deviation) of 4 runs with different random seeds.

### B.7.1 Additional Results on ZINC with Edge Features

In Table 12, we provide additional results on ZINC-12k with edge features. Namely, we reiterate Table 2 while also including R-GCN results for each encoding type, as well as RWSE+MoSE results where we concatenate the RWSE vector to the MoSE vector for a combined encoding. As stated in the main text, MoSE consistently outperforms RWSE. Interestingly, the comparison between MoSE

and the combination of RWSE+MoSE is less consistent. Although RWSE+MoSE contains more information than MoSE alone, there are practical limitations of the combined encoding which may explain its comparatively-worse performance on some models – e.g., the extra information provided by *two* positional/structural encodings requires drastic changes in the model hyperparameters, such as an increase in model width, which may degrade performance more than the additional encoding information improves it (especially if the two encodings contain redundant information). Regardless, the differences between MoSE and RWSE+MoSE are small across all tasks.

Table 12: We report mean absolute error (MAE) for graph regression on ZINC-12K (with edge features) across several models. MoSE yields substantial improvements over RWSE for every architecture. Best results are shown in red, second best in blue

| Model | Encoding Type | | | |
|---|---|---|---|---|
| | *none* | RWSE | MoSE | RWSE+MoSE |
| MLP-E | $0.606_{\pm 0.002}$ | $0.361_{\pm 0.010}$ | $0.347_{\pm 0.003}$ | $0.349_{\pm 0.003}$ |
| R-GCN | $0.413_{\pm 0.005}$ | $0.207_{\pm 0.007}$ | $0.197_{\pm 0.004}$ | $0.188_{\pm 0.005}$ |
| GIN-E | $0.243_{\pm 0.006}$ | $0.122_{\pm 0.003}$ | $0.118_{\pm 0.007}$ | $0.117_{\pm 0.005}$ |
| GIN-E+VN | $0.151_{\pm 0.006}$ | $0.085_{\pm 0.003}$ | $0.068_{\pm 0.004}$ | $0.064_{\pm 0.003}$ |
| GT-E | $0.195_{\pm 0.025}$ | $0.104_{\pm 0.025}$ | $0.089_{\pm 0.018}$ | $0.090_{\pm 0.005}$ |
| GPS | $0.119_{\pm 0.011}$ | $0.069_{\pm 0.001}$ | $0.062_{\pm 0.002}$ | $0.065_{\pm 0.002}$ |

**Note:** The GPS+RWSE result presented in Tables 12 and 2 is marginally different from the result presented in Table 1 (0.069 MAE versus 0.070 MAE, respectively). This discrepancy is due to a difference in GPS hyperparameters: the results in Table 12/2 follow our hyperparameter-search procedure given in Appendix B.7.3, whereas Table 1 pulls the reported GPS+RWSE result from Rampásek et al. (2022) and thus follows their hyperparameter procedure.

### B.7.2 ADDITIONAL RESULTS ON ZINC WITHOUT EDGE FEATURES

Dwivedi et al. (2023) also present a set of experiments that uses the ZINC-12k dataset without edge features. This is because there are a number of models that do not take edge features into account. Therefore, we also perform a set of experiments on ZINC that do not include edge features and report these results in Table 13. Similarly to our experiments with edge features, we see that MoSE consistently outperforms RWSE across all models, whereas the comparison between MoSE and RWSE+MoSE yields mixed results.

Table 13: We report additional MAE results for various models on ZINC-12k without edge features. Best results are shown in red, second best in blue

| Model | *none* | RWSE | MoSE | RWSE+MoSE |
|---|---|---|---|---|
| MLP | $0.663_{\pm 0.002}$ | $0.263_{\pm 0.006}$ | $0.218_{\pm 0.005}$ | $0.202_{\pm 0.001}$ |
| GIN | $0.294_{\pm 0.012}$ | $0.190_{\pm 0.004}$ | $0.158_{\pm 0.004}$ | $0.168_{\pm 0.005}$ |
| GT | $0.674_{\pm 0.001}$ | $0.217_{\pm 0.007}$ | $0.209_{\pm 0.020}$ | $0.184_{\pm 0.001}$ |
| GPS | $0.178_{\pm 0.016}$ | $0.116_{\pm 0.001}$ | $0.102_{\pm 0.001}$ | $0.105_{\pm 0.006}$ |

### B.7.3 ZINC HYPERPARAMETERS

We describe the hyperparameters and grid-searches used for our experiments on ZINC (Tables 1, 2, 3, and 12). Note that all models are restricted to a parameter budget of 500K. See Table 14 for a rough summary of the final hyperparameters.

Table 14: Hyperparameter summary for MoSE results on ZINC from Table 2.

| Model | Layers | Hidden Dim | Batch Size | Learning Rate | Epochs | MoSE dim | #Heads |
|---|---|---|---|---|---|---|---|
| MLP-E | 8 | 64 | 32 | 0.001 | 1200 | 28 | N/A |
| GIN-E | 4 | 110 | 128 | 0.001 | 1000 | 42 | N/A |
| GIN-E+VN | 4 | 110 | 32 | 0.001 | 1000 | 42 | N/A |
| GT-E | 10 | 64 | 32 | 0.001 | 2000 | 28 | 4 |
| GPS | 8 | 92-64 | 32 | 0.001 | 2000 | 42 | 4 |

**MLP-E**    We perform a grid-search on MLP-E for our ZINC results presented in Table 2. We first fix the following hyperparameters:

| | |
|---|---|
| *Batch Size*: 32 | *Optimizer*: Adam |
| *Weight Decay*: 0.001 | *Base LR*: 0.001 |
| *Max Epochs*: 1200 | *LR Scheduler*: Cosine |
| *LR Warmup Epochs*: 10 | |

Then, we search MLP-E+*none* on every combination of the following values, with the best values shown in **bold**:

$$\text{Node Encoder Layers: } \{\mathbf{2}, 4, 8\}$$

$$\text{Node Encoder Width: } \{\mathbf{64}, 128, 256\}$$

$$\text{Edge Encoder Layers: } \{2, 4, \mathbf{8}\}$$

$$\text{Edge Encoder Width: } \{32, 64, \mathbf{128}\}$$

$$\text{Graph Encoder Depth: } \{1, 2, \mathbf{4}\}$$

Here, the "graph encoder" is an MLP that follows sum-pooling and precedes the MLP prediction head described in Appendix B.3. The graph encoder width is the sum of the node encoder width and the edge encoder width. Using the best edge encoder and graph encoder from this first search, we then search MLP-E+RWSE and MLP-E+MoSE on all combinations of:

$$\text{Node Encoder Layers : } \{2, 4^*, \mathbf{8}\} \qquad \text{Node Encoder Width : } \{\mathbf{64}, 128^*, 256\}$$

where the best values for MoSE are shown in **bold** and the best values for RWSE have an asterisk*.

**R-GCN and GIN-E**    We do not conduct hyperparameter searches on our MPNN results on ZINC shown in Tables 2 and 12, opting instead to use the configurations from Jin et al. (2024). Size hyperparameters for R-GCN refer to the results in Table 12, and hyperparameters that are not specific to R-GCN or GIN-E are used by both models in our experiments. The configurations are as follows:

| | |
|---|---|
| *Batch Size*: 128 | *Optimizer*: Adam |
| *Max Epochs*: 1000 | *Base LR*: 0.001 |
| *LR Scheduler*: Reduce on plateau | *LR Warmup Epochs*: 50 |
| *LR Reduce Factor*: 0.5 | *LR Patience (min)*: 10 (1e-5) |
| *GIN-E Layers*: 4 | *GIN-E Width*: 110 |
| *R-GCN Layers*: 4 | *R-GCN Width*: 125 |

**GIN-E+VN**    Again, we do not conduct hyperparameter searches for GIN-E+VN on ZINC (Table 2). Our configuration is as follows:

| | |
|---|---|
| *Batch Size*: 32 | *Optimizer*: AdamW |
| *Max Epochs*: 1000 | *Base LR*: 0.001 |
| *LR Scheduler*: Cosine annealing | *LR Warmup Epochs*: 50 |
| *GIN-E Layers*: 4 | *GIN-E Width*: 110 |
| *Weight Decay*: 0.00001 | |

**GT-E**  We describe our hyperparameter search for ZINC GT-E results given in Table 2. The following hyperparameters are fixed:

| | |
|---|---|
| *Batch Size*: 32 | *Optimizer*: AdamW |
| *Weight Decay*: 1e-5 | *Base LR*: 0.001 |
| *Max Epochs*: 2000 | *LR Scheduler*: Cosine |
| *LR Warmup Epochs*: 50 | *Attention Dropout*: 0.5 |

For all feature injections (*none*, RWSE, MoSE), we search the following model sizes (formatted as a tuple where components indicate layers, width, and number of heads, respectively):

$$\{(8, 84, 4), (10, 64, 4), (4, 120, 8), (4 \rightarrow 6, 92 \rightarrow 64, 4)\}$$

where $(4 \rightarrow 6, 92 \rightarrow 64, 4)$ denotes a varied size model which has an initial width of 92 for 4 layers, and then (after being down-projected by a linear layer) has a final width of 64 for 6 more layers. The $(10, 64, 4)$ size model was best for all feature injections.

**GPS**  We describe our hyperparameter search for ZINC GPS results given in Table 2. The following hyperparameters remain fixed throughout:

| | |
|---|---|
| *MPNN Type*: GIN-E | *Attention Type*: Standard |
| *Batch Size*: 32 | *Optimizer*: AdamW |
| *Weight Decay*: 1e-5 | *Base LR*: 0.001 |
| *Max Epochs*: 2000 | *LR Scheduler*: Cosine |
| *LR Warmup Epochs*: 50 | *Attention Dropout*: 0.5 |

For all feature injections (*none*, RWSE, MoSE), we search the following models sizes (formatted as a tuple where components indicate layers, width, and number of heads, respectively):

$$\{(8, 76, 4), (10, 64, 4), (4, 104, 8), (2 \rightarrow 6, 92 \rightarrow 64, 4)\}$$

where $(2 \rightarrow 6, 92 \rightarrow 64, 4)$ denotes a varied size model which has an initial width of 92 for 2 layers, and then (after being down-projected by a linear layer) has a final width of 64 for 6 more layers. The $(8, 76, 4)$ model performed best for *none*-encoding, while $(2 \rightarrow 6, 92 \rightarrow 64, 4)$ was best for RWSE and MoSE.

**GRIT**  For the GRIT results presented in Table 3, we simply remove the RRWP node and node-pair encodings from the configuration used in Ma et al. (2023), and replace them with MoSE that is constructed from $\mathsf{Spasm}(C_7) \cup \mathsf{Spasm}(C_8)$. We summarize the hyperparameters as follows:

| | |
|---|---|
| *Layers*: 10 | *Width/Heads*: 64 / 8 |
| *Batch Size*: 32 | *Optimizer*: AdamW |
| *Weight Decay*: 1e-5 | *Base LR*: 0.001 |
| *Max Epochs*: 2000 | *LR Scheduler*: Cosine |
| *LR Warmup Epochs*: 50 | *Attention Dropout*: 0.2 |

**Other Experiments on ZINC**  Hyperparameter searches for RWSE+MoSE results in Table 12 are conducted identically to the searches for MoSE and RWSE detailed above. Searches for ZINC without edge features (Table 13) are conducted analogously. See the code repository for details.

### B.8  QM9 EXPERIMENTAL DETAILS & ADDITIONAL RESULTS

The QM9 dataset is another molecular dataset that consists of 130,831 graphs (Wu et al., 2018). They are split into 110,831 graphs for training, 10,000 for validation, and 10,000 for testing. Node features indicate the atom type and other additional atom features, such as its atomic number, the number of Hydrogens connected to it, etc. The edge features indicate bond type between two atoms. The task in this dataset is to predict 13 different quantum chemical properties, ranging from a molecule's dipole moment ($\mu$) to its free energy ($G$). For all experiments *except* those using the GT model (Table 17), we include the use of edge features. All GPS results in Table 4 are the mean ($\pm$ standard deviation)

of 5 runs with different random seeds, while results for other models are taken directly from their original works (Brockschmidt, 2020; Alon & Yahav, 2021; Abboud et al., 2022; Dimitrov et al., 2023). All appendix results in Tables 15, 16, and 17 (excluding those that are copied from Table 4) are the mean ($\pm$ standard deviation) of 3 runs with different random seeds.

### B.8.1 ADDITIONAL RESULTS ON QM9

Table 15 provides additional results for GPS on QM9 where we use the combined encoding MoSE+RWSE. Additionally, we provide QM9 experiments with MLP-E (Appendix B.2) and the GT model (Dwivedi & Bresson, 2020) in Table 16 and Table 17 respectively. Across all models, we see that MoSE typically outperforms RWSE. There are notable exceptions however, such as GT on the tasks U0, U, and H, where GT+RWSE does particularly well (Table 17). The comparison between MoSE alone and RWSE+MoSE consistently favors the combined encoding on MLP-E and GT (Tables 16 and 17), whereas the results on GPS are more mixed, with more tasks favoring MoSE (Table 15). In Table 16, we also experiment with MLP-E+Hom where Hom denotes the *un*-rooted homomorphism counts $\text{Hom}(\mathcal{G}, \cdot)$ with $\mathcal{G} = \text{Spasm}(C_7) \cup \text{Spasm}(C_8)$. We inject these counts as a graph-level feature by concatenating them to the learned graph representation prior to the final MLP prediction head. We see in Table 16 that the node-rooted MoSE consistently outperforms its graph-level counterpart.

Table 15: We report MAE results for GPS with various encodings on the QM9 dataset.

| Property | GPS | +RWSE | +MoSE | +RWSE+MoSE |
|---|---|---|---|---|
| mu | $1.52_{\pm 0.02}$ | $1.47_{\pm 0.02}$ | $1.43_{\pm 0.02}$ | $1.45_{\pm 0.01}$ |
| alpha | $2.62_{\pm 0.38}$ | $1.52_{\pm 0.27}$ | $1.48_{\pm 0.16}$ | $1.72_{\pm 0.11}$ |
| HOMO | $1.17_{\pm 0.41}$ | $0.91_{\pm 0.01}$ | $0.91_{\pm 0.01}$ | $0.92_{\pm 0.01}$ |
| LUMO | $0.92_{\pm 0.01}$ | $0.90_{\pm 0.06}$ | $0.86_{\pm 0.01}$ | $0.88_{\pm 0.16}$ |
| gap | $1.46_{\pm 0.02}$ | $1.47_{\pm 0.02}$ | $1.45_{\pm 0.02}$ | $1.48_{\pm 0.01}$ |
| R2 | $6.82_{\pm 0.31}$ | $6.11_{\pm 0.16}$ | $6.22_{\pm 0.19}$ | $6.01_{\pm 0.03}$ |
| ZPVE | $2.25_{\pm 0.18}$ | $2.63_{\pm 0.44}$ | $2.43_{\pm 0.27}$ | $2.23_{\pm 0.25}$ |
| U0 | $0.96_{\pm 0.34}$ | $0.83_{\pm 0.17}$ | $0.85_{\pm 0.08}$ | $0.80_{\pm 0.05}$ |
| U | $0.81_{\pm 0.05}$ | $0.83_{\pm 0.15}$ | $0.75_{\pm 0.03}$ | $0.78_{\pm 0.03}$ |
| H | $0.81_{\pm 0.26}$ | $0.86_{\pm 0.15}$ | $0.83_{\pm 0.09}$ | $0.87_{\pm 0.16}$ |
| G | $0.77_{\pm 0.04}$ | $0.83_{\pm 0.12}$ | $0.80_{\pm 0.14}$ | $0.72_{\pm 0.03}$ |
| Cv | $2.56_{\pm 0.72}$ | $1.25_{\pm 0.05}$ | $1.02_{\pm 0.04}$ | $1.03_{\pm 0.08}$ |
| Omega | $0.40_{\pm 0.01}$ | $0.39_{\pm 0.02}$ | $0.38_{\pm 0.01}$ | $0.39_{\pm 0.01}$ |

Table 16: We report MAE results for MLP-E with various encodings on the QM9 dataset.

| Property | MLP-E | +RWSE | +MoSE | +Hom | +RWSE+MoSE |
|---|---|---|---|---|---|
| mu | $5.31_{\pm 0.03}$ | $4.91_{\pm 0.06}$ | $4.88_{\pm 0.05}$ | $5.21_{\pm 0.04}$ | $4.84_{\pm 0.03}$ |
| alpha | $6.76_{\pm 0.04}$ | $5.08_{\pm 0.18}$ | $5.11_{\pm 0.11}$ | $5.52_{\pm 0.06}$ | $4.83_{\pm 0.09}$ |
| HOMO | $2.96_{\pm 0.03}$ | $2.46_{\pm 0.03}$ | $2.36_{\pm 0.04}$ | $2.75_{\pm 0.01}$ | $2.34_{\pm 0.02}$ |
| LUMO | $3.17_{\pm 0.04}$ | $2.76_{\pm 0.3}$ | $2.65_{\pm 0.03}$ | $3.11_{\pm 0.01}$ | $2.57_{\pm 0.03}$ |
| gap | $4.34_{\pm 0.02}$ | $3.58_{\pm 0.02}$ | $3.45_{\pm 0.05}$ | $4.08_{\pm 0.07}$ | $3.35_{\pm 0.04}$ |
| R2 | $43.69_{\pm 1.08}$ | $26.69_{\pm 0.31}$ | $26.50_{\pm 0.43}$ | $35.85_{\pm 0.45}$ | $24.97_{\pm 0.45}$ |
| ZPVE | $14.36_{\pm 0.60}$ | $10.42_{\pm 1.24}$ | $8.94_{\pm 0.83}$ | $11.51_{\pm 2.33}$ | $10.28_{\pm 0.35}$ |
| U0 | $8.06_{\pm 0.78}$ | $5.39_{\pm 0.58}$ | $5.30_{\pm 0.41}$ | $5.45_{\pm 0.51}$ | $4.74_{\pm 0.28}$ |
| U | $8.33_{\pm 0.56}$ | $5.34_{\pm 0.59}$ | $5.19_{\pm 0.39}$ | $5.17_{\pm 0.86}$ | $5.10_{\pm 0.62}$ |
| H | $8.19_{\pm 0.33}$ | $5.17_{\pm 0.28}$ | $4.77_{\pm 0.16}$ | $5.49_{\pm 0.38}$ | $5.08_{\pm 0.33}$ |
| G | $7.58_{\pm 0.14}$ | $5.83_{\pm 0.46}$ | $5.28_{\pm 0.54}$ | $5.04_{\pm 0.28}$ | $5.03_{\pm 0.36}$ |
| Cv | $6.06_{\pm 0.14}$ | $3.80_{\pm 0.15}$ | $3.73_{\pm 0.25}$ | $4.19_{\pm 0.16}$ | $3.72_{\pm 0.44}$ |
| Omega | $1.61_{\pm 0.04}$ | $1.29_{\pm 0.02}$ | $1.17_{\pm 0.02}$ | $1.49_{\pm 0.02}$ | $1.24_{\pm 0.05}$ |

Table 17: We report MAE results for GT with various encodings on the QM9 dataset.

| Property | GT | +RWSE | +MoSE | +RWSE+MoSE |
|---|---|---|---|---|
| mu | $4.12_{\pm0.05}$ | $2.53_{\pm0.06}$ | $2.43_{\pm0.01}$ | $2.31_{\pm0.06}$ |
| alpha | $8.80_{\pm4.65}$ | $1.43_{\pm0.03}$ | $2.00_{\pm0.08}$ | $1.32_{\pm0.01}$ |
| HOMO | $1.72_{\pm0.01}$ | $1.15_{\pm0.01}$ | $1.11_{\pm0.01}$ | $1.08_{\pm0.01}$ |
| LUMO | $1.63_{\pm0.01}$ | $1.05_{\pm0.02}$ | $1.02_{\pm0.01}$ | $0.99_{\pm0.01}$ |
| gap | $2.38_{\pm0.03}$ | $1.67_{\pm0.01}$ | $1.62_{\pm0.01}$ | $1.57_{\pm0.02}$ |
| R2 | $26.73_{\pm8.59}$ | $8.30_{\pm0.32}$ | $10.69_{\pm1.56}$ | $7.80_{\pm0.17}$ |
| ZPVE | $10.39_{\pm1.35}$ | $2.46_{\pm0.04}$ | $4.66_{\pm2.09}$ | $2.37_{\pm0.03}$ |
| U0 | $6.27_{\pm1.95}$ | $0.94_{\pm0.10}$ | $2.56_{\pm1.66}$ | $0.98_{\pm0.57}$ |
| U | $5.38_{\pm0.17}$ | $0.92_{\pm0.03}$ | $3.10_{\pm2.75}$ | $1.40_{\pm0.72}$ |
| H | $5.30_{\pm0.17}$ | $0.92_{\pm0.06}$ | $3.49_{\pm2.02}$ | $0.96_{\pm0.51}$ |
| G | $5.44_{\pm0.51}$ | $0.88_{\pm0.08}$ | $2.72_{\pm1.21}$ | $0.63_{\pm0.03}$ |
| Cv | $4.32_{\pm0.07}$ | $1.25_{\pm0.01}$ | $1.43_{\pm0.04}$ | $1.15_{\pm0.01}$ |
| Omega | $1.25_{\pm0.01}$ | $0.50_{\pm0.01}$ | $0.47_{\pm0.02}$ | $0.46_{\pm0.01}$ |

### B.8.2 QM9 HYPERPARAMETERS

We give a rough summary of the hyperparameters that we use for our QM9 experiments in Table 21, and we describe our search procedures below. Different tasks use different hyperparameters. The precise setup can be found in our code repository.

**MLP-E**  We do not perform any grid searching for MLP-E on QM9 (Table 16) and simply use the same hyperparameters for every task and every encoding method, given below:

| | | |
|---|---|---|
| *Node Encoder Depth*: 4 | *Node Encoder Width*: 220 | *Edge Encoder Depth*: 2 |
| *Edge Encoder Width*: 32 | *Graph Encoder Depth*: 3 | *Graph Encoder Width*: 260 |
| *Batch Size*: 128 | *Optimizer*: Adam | *Weight Decay*: 0.001 |
| *Base LR*: 0.001 | *Max Epochs*: 800 | *LR Scheduler*: Cosine |
| *LR Warmup Epochs*: 10 | *Dropout*: 0.1 | |

**GPS**  We perform a 2-round hyperparameter search for GPS results on QM9 (Table 4 & 15). Firstly, the following default hyperparameters remain constant throughout:

| | |
|---|---|
| *Batch Size*: 128 | *Optimizer*: AdamW |
| *Weight Decay*: 1e-5 | *Base LR*: 0.001 |
| *Max Epochs*: 1200 | *LR Scheduler*: Cosine |
| *LR Warmup Epochs*: 20 | *Attention Dropout*: 0.5 |

We grid-search each encoding method independently (on a reduced 700 epochs). The first round only includes the tasks R2, ZPVE, U0, and gap, and we search over tuples of the form MPNN-type $\times$ depth $\times$ width $\times$ heads:

$$\Big( \{\text{R-GCN}, \text{GIN-E}\} \times \{(8, 128, 4), (10, 64, 4)\} \Big) \cup \Big( \{(\text{GIN-E}, 12, 256, 8)\} \Big)$$

One should interpret the $\times$ symbol as the Cartesian product on sets. Then, the set above describes the tuples (and thus the model configurations) that we search over. After performing this search, the best two models for each task are subsequently searched on every task/encoding according to the task groups given in Table 20. These task groups are formed according to notions of task similarity described by Gilmer et al. (2017).

For example, the best performing GPS+MoSE model on the gap task in the first round of our search is (R-GCN, 10, 64, 4), and the second best model is (GIN-E, 8, 128, 4). Since the LUMO task is in gap's group (as per Table 20), we proceed to search these two GPS+MoSE models, (R-GCN, 10, 64, 4) and (GIN-E, 8, 128, 4), on the LUMO task. This second round of searching reveals the final configuration for GPS+MoSE on LUMO, which turns out to be (R-GCN, 10, 64, 4). We repeat this procedure for every task-encoding pair, and we end up with the model configurations given in Table 18

**GT**  For our GT results on QM9 (Table 17), we perform the same 2-round search procedure as with GPS, starting from the same default configuration (except we use 700 epoch for our final runs instead of 1200). Our first round searches over tuples in depth $\times$ width $\times$ heads $\times$ base LR:

$$\{(8, 128, 8), (10, 64, 4), (12, 256, 8)\} \times \{0.0001, 0.00001\}$$

Our second round proceeds analogously to that described in the GPS search (again using Table 20), but highly varied training behavior across the tasks requires us to perform some additional (minor) task-specific tuning in the second round. We omit the details of this tuning here, asking readers to refer to the code repository. Ultimately, we get the final model configurations shown in Table 19.

Table 18: The final GPS configurations for QM9 results.

|        | None              | RWSE              | MoSE              | RWSE+MoSE         |
|--------|-------------------|-------------------|-------------------|-------------------|
| mu     | (R-GCN, 10, 64)   | (R-GCN, 10, 64)   | (R-GCN, 10, 64)   | (R-GCN, 10, 64)   |
| alpha  | (R-GCN, 10, 64)   | (R-GCN, 10, 64)   | (R-GCN, 10, 64)   | (R-GCN, 8, 128)   |
| HOMO   | (GIN-E, 8, 128)   | (R-GCN, 10, 64)   | (R-GCN, 10, 64)   | (R-GCN, 10, 64)   |
| LUMO   | (R-GCN, 10, 64)   | (R-GCN, 10, 64)   | (R-GCN, 10, 64)   | (R-GCN, 10, 64)   |
| gap    | (R-GCN, 10, 64)   | (R-GCN, 10, 64)   | (R-GCN, 10, 64)   | (R-GCN, 10, 64)   |
| R2     | (R-GCN, 10, 64)   | (R-GCN, 8, 128)   | (R-GCN, 10, 64)   | (R-GCN, 8, 128)   |
| ZPVE   | (GIN-E, 10, 64)   | (GIN-E, 8, 128)   | (GIN-E, 8, 128)   | (GIN-E, 10, 64)   |
| U0     | (GIN-E, 10, 64)   | (GIN-E, 8, 128)   | (GIN-E, 8, 128)   | (R-GCN, 10, 64)   |
| U      | (R-GCN, 10, 64)   | (GIN-E, 10, 64)   | (GIN-E, 8, 128)   | (GIN-E, 10, 64)   |
| H      | (GIN-E, 10, 64)   | (GIN-E, 10, 64)   | (GIN-E, 8, 128)   | (GIN-E, 10, 64)   |
| G      | (R-GCN, 10, 64)   | (GIN-E, 10, 64)   | (GIN-E, 8, 128)   | (GIN-E, 10, 64)   |
| Cv     | (R-GCN, 10, 64)   | (GIN-E, 10, 64)   | (R-GCN, 10, 64)   | (R-GCN, 10, 64)   |
| Omega  | (GIN-E, 10, 64)   | (GIN-E, 10, 64)   | (R-GCN, 10, 64)   | (R-GCN, 10, 64)   |

*Tuples are given as (MPNN-type, depth, width). All of our final models use 4 attention heads.

Table 19: The final GT configurations for QM9 results.

|        | None                         | RWSE                        | MoSE                        | RWSE+MoSE                   |
|--------|------------------------------|-----------------------------|-----------------------------|-----------------------------|
| mu     | $(8, 128, 10^{-4}, 0)$       | $(12, 256, 10^{-4}, 0)$     | $(12, 256, 10^{-4}, 0)$     | $(12, 256, 10^{-4}, 0)$     |
| alpha  | $(8, 128, 10^{-4}, 0)$       | $(12, 256, 10^{-4}, 0)$     | $(8, 128, 10^{-4}, 0)$      | $(12, 256, 10^{-4}, 0)$     |
| HOMO   | $(12, 256, 10^{-4}, 0)$      | $(12, 256, 10^{-4}, 0)$     | $(12, 256, 10^{-4}, 0)$     | $(12, 256, 10^{-4}, 0)$     |
| LUMO   | $(12, 256, 10^{-4}, 0)$      | $(12, 256, 10^{-4}, 0)$     | $(12, 256, 10^{-4}, 0)$     | $(12, 256, 10^{-4}, 0)$     |
| gap    | $(12, 256, 10^{-4}, 0)$      | $(12, 256, 10^{-4}, 0)$     | $(12, 256, 10^{-4}, 0)$     | $(12, 256, 10^{-4}, 0)$     |
| R2     | $(10, 64, 10^{-3}, 0)$       | $(12, 256, 10^{-4}, 0)$     | $(12, 256, 10^{-4}, 0)$     | $(12, 256, 10^{-4}, 0)$     |
| ZPVE   | $(8, 48, 10^{-4}, 0.3)$      | $(8, 128, 10^{-4}, 0)$      | $(8, 128, 10^{-4}, 0)$      | $(8, 128, 10^{-4}, 0)$      |
| U0     | $(10, 64, 10^{-4}, 0)$       | $(8, 128, 10^{-4}, 0)$      | $(8, 128, 10^{-4}, 0)$      | $(12, 256, 10^{-4}, 0)$     |
| U      | $(12, 256, 10^{-4}, 0)$      | $(8, 128, 10^{-4}, 0)$      | $(8, 128, 10^{-4}, 0)$      | $(12, 256, 10^{-4}, 0)$     |
| H      | $(10, 64, 10^{-4}, 0.5)$     | $(8, 128, 10^{-4}, 0)$      | $(10, 64, 10^{-4}, 0)$      | $(12, 256, 10^{-4}, 0)$     |
| G      | $(8, 48, 10^{-4}, 0.3)$      | $(8, 128, 10^{-4}, 0)$      | $(10, 64, 10^{-4}, 0)$      | $(12, 256, 10^{-4}, 0)$     |
| Cv     | $(8, 48, 10^{-4}, 0.3)$      | $(8, 128, 10^{-4}, 0)$      | $(8, 128, 10^{-4}, 0)$      | $(8, 128, 10^{-4}, 0)$      |
| Omega  | $(8, 48, 10^{-4}, 0.3)$      | $(8, 128, 10^{-4}, 0)$      | $(8, 128, 10^{-4}, 0)$      | $(8, 128, 10^{-4}, 0)$      |

*Tuples are given as (depth, width, base-LR, dropout), where dropout is applied to the feed-forward modules interweaving attention layers, and is tuned in the task-specific portion of the second round. The number of heads can be inferred from the depth/width.

Table 20: Task groups for QM9 grid-searching procedure.

| 1st Round | 2nd Round |
|---|---|
| R2 | R2, mu, alpha |
| ZPVE | ZPVE, Omega, Cv |
| U0 | U0, H, G, U |
| gap | gap, HOMO, LUMO |

Table 21: High-level summary of hyperparameters used for our QM9 experiments.

| Model | Layers | Hidden Dim | Batch Size | Learning Rate | Epochs | #Heads |
|---|---|---|---|---|---|---|
| MLP-E | 4 | 220 | 128 | 0.001 | 800 | N/A |
| GT | 8, 10, 12 | 48, 64, 128, 256 | 128 | 0.001, 0.0001 | 700 | 4, 8 |
| GPS | 8, 10 | 64, 128 | 128 | 0.001 | 1200 | 4 |

### B.9 SYNTHETIC DATASET EXPERIMENTAL DETAILS

We generate a new synthetic dataset where the goal is to predict the fractional domination number of a graph. The dataset contains 10,000 randomly-generated Erdős-Rényi graphs, where edge density varies between $[0.25, 0.75]$ and the number of nodes varies between $[16, 32]$. There are *no* initial node features or edge features provided, so the model must infer the target using only graph structure.

A fractional dominating function of a graph $G$ is a weight assignment $\alpha : V(G) \to [0, 1]$ such that for every vertex $v$, the sum of the weights assigned to $v$ and its neighbors is at least 1. The total weight of $\alpha$ is given by $\sum_{v \in V(G)} \alpha(v)$. The *fractional domination number* of $G$ is the smallest total weight of any fractional dominating function on $G$ (Scheinerman & Ullman, 2013).

We select a 4-layer MLP and a simple Graph Transformer (Dwivedi & Bresson, 2020) for our experiments in order to focus solely on the power of the graph inductive biases (MoSE, RWSE, LapPE). That is, MLP and GT do not receive the adjacency matrix, initial node features, or initial edge features as an input: our model must infer the fractional domination number using *only* the node-level positional/structural encoding vectors provided.

In Table 5, we report the mean ($\pm$ standard deviation) of 4 runs with different random seeds. For each model, we do not perform any hyperparameter tuning. In order to control for the large homomorphism count values generated by MoSE on this dataset, we scale the raw counts down by taking the $\log_{10}$ of the true values. Please refer to the code repository for specific configurations and data files.

**MLP** Our MLP experiments on the synthetic dataset (Table 5) use the following hyperparameters for all encodings:

| | | |
|---|---|---|
| *Depth*: 4 | *Width*: 145 | *Global Pooling*: Mean |
| *Batch Size*: 128 | *Optimizer*: Adam | *Weight Decay*: 0.0 |
| *Base LR*: 0.001 | *Max Epochs*: 1000 | *LR Scheduler*: Reduce on Plateau |
| *LR Reduce Factor*: 0.5 | *Dropout*: 0.2 | *LR Patience (min): 10 (1e-5)* |

**GT** Our GT experiments on the synthetic dataset (Table 5) use the following hyperparameters for all encodings:

| | | |
|---|---|---|
| *Depth*: 10 | *Width*: 64 | *Heads*: 4 |
| *Batch Size*: 128 | *Optimizer*: AdamW | *Weight Decay*: 0.001 |
| *Base LR*: 0.0001 | *Max Epochs*: 1200 | *LR Scheduler*: Cosine |
| *LR Warmup Epochs*: 30 | *Attention Dropout*: 0.5 | |

