# OpenReview forum: "Homomorphism Counts as Structural Encodings for Graph Learning"
_ICLR.cc/2025/Conference — ICLR 2025 Poster_

### Official Review · Reviewer_aNPU · 2024-10-28

**Soundness:** 4
**Presentation:** 3
**Contribution:** 3
**Rating:** 6
**Confidence:** 4

**Summary:**

Graph Transformers apply self-attention on graph nodes and to capture the graph structure, they use positional or structural encodings. This paper proposes to use homomorphism count vectors as the employed structural encoding. The authors show that homomorphism counts are more expressive than the commonly used random walk structural encoding. The proposed encoding is evaluated on two molecular property prediction datasets (ZINC-12k and QM9), and also on a synthetic dataset. The results demonstrate that the proposed encoding outperforms other well-known positional and structural encodings across a range of architectures.

**Strengths:**

- The proposed structural encoding seems to better capture the graph structure than other enocodings since it outperforms the baseline encodings (including the commonly used random walk structural encoding) on all considered datasets.

- Besides the empirical results, the paper makes a substantial theoretical contribution since it is shown that the proposed encoding is strictly more expressive than the random walk structural encoding.

- The paper is dense in some parts, but overall it is very well-written and the presentation is clear.

**Weaknesses:**

- Even though the proposed encoding outperforms the random walk structural encoding, the reported results are not very convincing. On most datasets, the difference in performance between the random walk structural encoding and the proposed encoding is very small. On QM9, the random walk structural encoding even outperforms the proposed encoding for some targets. It is thus unclear whether there is indeed a need for a new encoding in this setting since it does not lead to significance performance gains over the existing method.

- The complexity of computing the proposed encoding is not discussed in the paper, but I would expect that the time required to compute this encoding is significantly higher than the one required to compute the random walk structural encoding. In my understanding, the running time depends on the pattern graph family $\mathcal{G}$. I would thus suggest the authors report in the paper the pre-processing time for each one of the considered encodings. If the additional computational cost is high and given the marginal performance improvements it offers, why one would choose the proposed encoding over other encodings?

- One of my main concerns with this paper is regarding the novelty of the work. Clearly this is not the first paper to use homomorphism counts in the field of graph machine learning. Several previous works which are discussed in section 2 in the paper have annotated nodes with homomorphism counts. The main difference between those works and this paper is that the learning models employed in this paper are Graph Transformers and not MPNNs. However, I doubt if this in enough for the paper to be considered novel.

- The authors choose to integrate the proposed encoding into Graph Transformers, but all the theoretical results presented in the paper hold for the encoding itself (no assumptions are made about the underlying model). Thus, Graph Transformers could be replaced with MPNNs, and all the results would still hold. I thus wonder why the authors chose to place that much focus of the paper on Graph Transformers.

**Questions:**

- The proposed method relies on a finite set of fixed pattern graphs. One needs to choose the types of considered pattern graphs and this usually requires some domain knowledge. In case such knowledge is not available, what kind of patterns should be chosen?

---

> ### Author Response · Authors · 2024-11-21
>
> We thank the Reviewer for the feedback and address their main concerns below:
> >It is thus unclear whether there is indeed a need for a new encoding in this setting since it does not lead to significance performance gains over the existing method.
>
> **Answer:**  We would like to emphasize that MoSE **strictly generalizes** RWSE. For practical applications, addressing task-specific use cases is important and MoSE is flexible enough to accommodate these changes, whereas other encoding methods (i.e. RWSE) cannot.
>
> We provide an additional experiment where we evaluate the self-attention Graph Transformer (Dwivedi & Bresson, 2020) on the peptides-struct and peptides-func datasets to isolate the performance gain of MoSE vs RWSE. The empirical results support the benefits of MoSE which leads to substantial improvements over RWSE (please see our global response for the specific results).
>
> Furthermore, we extend the ablation study of positional/structural encodings from GPS (Rampasek et al., 2022) with the CIFAR10 dataset, which again highlights the benefits of MoSE across different domains
>
> >I would thus suggest the authors report in the paper the pre-processing time for each one of the considered encodings. If the additional computational cost is high and given the marginal performance improvements it offers, why one would choose the proposed encoding over other encodings?
>
> **Answer:** Computing the homomorphism counts for MoSE is highly parallelizable since the homomorphism from each pattern can be counted independently from one another. Empirically, it took only **~0.02 sec** to compute MoSE for a given graph, whereas training various Transformer models on datasets such as QM9 took as long as **15-30 hours** for a single run. Therefore, the time to compute MoSE is almost negligible overall. We thank the reviewer for this suggestion and will report the pre-processing time in our revised version.
>
> Since MoSE generalizes popular encodings, we see it also as a way to trade off more compute to achieve a more informative (and thus ideally better performing and generalizing) encoding. This kind of adaptability is not possible in RWSE or other popular encodings.
>
> >The main difference between those works and this paper is that the learning models employed in this paper are Graph Transformers and not MPNNs. However, I doubt if this in enough for the paper to be considered novel.
>
> **Answer:** We specifically study the power of homomorphism counts as a graph inductive bias in their own right, and we prove that MoSE strictly generalizes RWSE, which is a very popular encoding technique. Although our work is most closely related to Jin et al. 2024, their focus is on identifying which motif parameters should be used to reach a desired level of expressivity, whereas our focus is to study the expressivity of the homomorphism counts as a structural encoding technique themselves in comparison with existing encodings.
>
> >The authors choose to integrate the proposed encoding into Graph Transformers, but all the theoretical results presented in the paper hold for the encoding itself (no assumptions are made about the underlying model). Thus, Graph Transformers could be replaced with MPNNs, and all the results would still hold. I thus wonder why the authors chose to place that much focus of the paper on Graph Transformers.
>
> **Answer:** Indeed, our results are actually more general than just for Graph Transformers. We focus our empirical results around Transformers because these models have been shown to work well empirically, and we wanted to isolate the performance of the encodings alone, without any confounding graph inductive bias being provided by message passing in MPNNs.

---

> ### Author Response · Authors · 2024-11-21
>
> >The proposed method relies on a finite set of fixed pattern graphs. One needs to choose the types of considered pattern graphs and this usually requires some domain knowledge. In case such knowledge is not available, what kind of patterns should be chosen?
>
> **Answer:** We thank the reviewer for raising this point, and we highlight our approach for each of these cases:
>
> 1. **Domain knowledge available:** In this scenario, MoSE can be adapted to capture domain-specific patterns and guarantee a precise level of expressivity with respect to these patterns. This flexibility is not possible in existing encoding techniques, where certain patterns cannot be captured no matter how the encoding is parameterized.
> 2. **No domain knowledge available:** In this scenario, there is no reference point for expressivity that needs to be reached. Any finite set of features will fail to express some (infinitely many) functions. This makes every finite method equally bad in terms of achieving the right expressivity by pure luck when absolutely no knowledge is available. The framework of graph motif parameters in fact nicely determines which functions are not expressed by any finite set of homomorphism counts. In this case, we use all connected graphs up to $k$ vertices, since this is arguably the most neutral approach.
>
>
> References:
>
> Dwivedi & Bresson. A generalization of transformer networks to graphs. arXiv preprint arXiv:2012.09699, 2020.
>
> Rampasek et al. Recipe for a general, powerful, scalable graph transformer. In *NeurIPS*, 2022.
>
> Jin et al. Homomorphism Counts for Graph Neural Networks: All About That Basis. In *ICML*, 2024.

---

> > ### Comment · Reviewer_aNPU · 2024-11-23
> >
> > Thank you to the authors for the feedback. While the theoretical contribution of the work is significant, I still have some concerns about the empirical performance of the proposed encodings and its practical benefit:
> >
> > - In the empirical evaluation, the proposed encoding does not seem to bring any major improvements over the random walk structural encoding. It is thus not clear whether there is a need for a new encoding. In addition, on Peptides-struct the proposed encoding is significantly outperformed by SAN+RWSE and Transformer+LapPE (results reported in the paper that introduced the Peptides dataset [1]), while on Peptides-func its performance is comparable with that of the two aforementioned models.
> >
> > - From the practitioner's point of view, the proposed encoding is much more complex than the random walk structural encoding since some patterns need to be chosen. In case no domain knowledge available, the authors propose to use all connected graphs up to k vertices. This could incur significant additional computational costs if the input graphs are large and dense even though computation can be parallelized.
> >
> > - The expressivity of the homomorphism counts has already been studied in [2] and [3]. I would suggest the authors better explain in the manuscript how this work is different from the above two studies.
> >
> > [1] Dwivedi, V. P., Rampášek, L., Galkin, M., Parviz, A., Wolf, G., Luu, A. T., & Beaini, D. Long range graph benchmark. In NeurIPS'22, pp. 22326-22340.\
> > [2] Barceló, P., Geerts, F., Reutter, J., & Ryschkov, M. Graph neural networks with local graph parameters. In NeurIPS'21, pp. 25280-25293.\
> > [3] Lanzinger, M., & Barceló, P. On the power of the Weisfeiler-Leman test for graph motif parameters. In ICLR'24.

---

> ### Author Response · Authors · 2024-11-24
>
> We thank the reviewer for their reply and their acknowledgment of the significant theoretical contribution of our work. We further address the outstanding concerns below:
>
> >In the empirical evaluation, the proposed encoding does not seem to bring any major improvements over the random walk structural encoding. It is thus not clear whether there is a need for a new encoding.
>
> MoSE is an expressive encoding method that has explicit theoretical guarantees. Therefore, MoSE is a very strong alternative **when the expressive power of RWSE is insufficient**. For instance, in Figure 1, we show that RWSE cannot distinguish between two functional groups within a molecule. If the task at hand were to rely on such functional groups, our theoretical results show that RWSE cannot capture this information. This scenario is not unlikely in the real world, since functional groups play a key role in determining a molecule’s reactivity within certain settings. Practically speaking, MoSE is flexible enough to be adapted to domain-specific tasks, which is clearly not the case for RWSE. This finding is also supported experimentally on diverse datasets.
>
> >In addition, on Peptides-struct the proposed encoding is significantly outperformed by SAN+RWSE and Transformer+LapPE (results reported in the paper that introduced the Peptides dataset [1]), while on Peptides-func its performance is comparable with that of the two aforementioned models.
>
> SAN (Kreuzer et al., 2021) is fundamentally a different model from GT (Dwivedi & Bresson, 2020), which accounts for the difference in performance. For a more fair comparison of RWSE vs MoSE, we refer the reviewer to our global response, where we present a direct comparison of GT+RWSE vs GT+MoSE. We select GT because it provides a “pure” comparison of RWSE vs MoSE as a graph inductive bias, since the standard self-attention GT model does not contain additional confounding graph inductive biases that other models may have (e.g. SAN incorporates Laplacian spectral information as well). It is a important future avenue to study the combination of different inductive biases and our paper will hopefully play a crucial role in those investigations.
>
> >From a practitioner's point of view, the proposed encoding is much more complex than the random walk structural encoding since some patterns need to be chosen. In case no domain knowledge available, the authors propose to use all connected graphs up to k vertices. This could incur significant additional computational costs if the input graphs are large and dense even though computation can be parallelized.
>
> **MoSE can distinguish all graphs that RWSE distinguishes** by recovering the exact values of the RWSE vector by counting cycle homomorphisms with reciprocal degree weighting (detailed in the appendix).  As such, RWSE is no more than a special case of MoSE that chooses cycles to be the patterns of interest. If the baseline performance of RWSE is a sufficient starting point, one can simply construct MoSE with cycles (to recover RWSE), and any additional “complexity” for capturing additional patterns can be done on a flexible task-specific basis. Furthermore, from a purely computational perspective, the bottleneck in training for large and dense graphs is due to the quadratic runtime complexity of Graph Transformers rather than the computation of the homomorphism counts for small $k$ (=size of the motifs).
>
> >The expressivity of the homomorphism counts has already been studied in [2] and [3]. I would suggest the authors better explain in the manuscript how this work is different from the above two studies.
>
> The primary difference is that we study the strength of homomorphism counts in terms of positional/structural encoding. This is different than studying their power with MPNNs which themselves already include inductive bias. Our analysis also entails results which are not implied by any either [2,3], such as MoSE>RWSE, the associated limitations of RWSE, and how the use of MoSE in GTs compares to the expressive power of MPNNs. We thank the reviewer for their suggestion and will be sure to elaborate this further in the revised version of our paper.
>
> References:
>
> Kreuzer et al. Rethinking Graph Transformers with Spectral Attention. In *NeurIPS*, 2021.
>
> Dwivedi & Bresson. A generalization of transformer networks to graphs. arXiv preprint arXiv:2012.09699, 2020.

---

> > ### Comment · Reviewer_aNPU · 2024-11-25
> >
> > Thanks a lot for the response. Even though I am not fully convinced of MoSE's practical value, the response has partially addressed my concerns and I will raise my score to a 6.
> >
> > I would suggest the authors construct a dataset (where molecules are classified based on their functional groups, but even a synthetic dataset would suffice) to also empirically demonstrate that GT+MoSE can significantly outperform GT+RWSE in some task.
> >
> > I understand that for a fair comparison the authors use GT+RWSE as a baseline on the Peptides datasets. However, in my view it is important for the user to know whether the proposed encoding (combined with SAN or some other model) can outperform standard approaches such as SAN+RWSE.

---

> > > ### Author Response · Authors · 2024-11-26
> > >
> > > We thank the reviewer for all their feedback and for raising their score based on our rebuttal. We will aim to construct the suggested synthetic dataset and include the additional results in the revised version of our paper.

---

### Official Review · Reviewer_7PEY · 2024-10-31

**Soundness:** 3
**Presentation:** 3
**Contribution:** 3
**Rating:** 6
**Confidence:** 5

**Summary:**

The paper introduces a novel positional encoding (PE) for graph transformers based on homomorphism counts and explains it's expressivity in terms of k-WL and the comparison to the random-walk positional encoding. The model is evaluated with two different graph transformers as base models and, over molecule and synthetic data, it shows some improvements.

**Strengths:**

- The research topic makes sense given the recent insights of using homomorphisms in MPNNs.
- The paper is well-structured
- The synthetic data experiment is interesting, and the authors consider two different transformers as base models
- The model seems to achieve good improvements generally (the number of evidence seems a bit low)

**Weaknesses:**

- The paper is sometimes annoying to read:
     - The reader is assumed to know quite some details (e.g.,
W/o knowing the details of RWSE, you don't get the introductory example; the tables mention models like GSN and CIN, which are never explained; oblivious k-WL; l.232 how does GRIT deviate from this? What is Spasm(C7∪C8)?)
     - The expressivity analysis is interesting but it would have been nice if the papers would have provided some ideas about the proofs, to better get to know the nature of the PE.

- With all the molecular datasets considered, it would have been nice if the evaluation contained some baseline from the chemistry domain, to see how the model compares in realistic settings.
- Regarding the QM9 experiment, I am computer scientist myself and cannot entirely judge the validity, but wanted to point out that some chemists have strong doubts about this kind of experiment:
"While I appreciate that some claim a 1D representation like SMILES can capture some component of a 3D structure, predicting a property of one molecular conformation from a SMILES doesn’t make much sense. "
https://practicalcheminformatics.blogspot.com/2023/08/we-need-better-benchmarks-for-machine.html

- Beyond that, there's only Zinc12k, which is a bit of a low number of datasets for evaluation

Smaller comments:
- typo: alternative olibious k-WL
- l. 296 "f a graph in G." Is this \mathcal{G} ?
- FYI There's a recent related work: Deep Homomorphism Networks
https://neurips.cc/virtual/2024/poster/95659

**Questions:**

Maybe the authors can point out why they evaluate on only these datasets.

---

> ### Author Response · Authors · 2024-11-21
>
> We thank the reviewer for the constructive feedback and respond to the main points below:
> >The reader is assumed to know quite some details
>
> **Answer:** We thank the reviewer for raising these points and will provide a more detailed explanation of our notion and discussion in the revised paper.
>
> >With all the molecular datasets considered, it would have been nice if the evaluation contained some baseline from the chemistry domain, to see how the model compares in realistic settings.
>
> **Answer:** Our approach is general in that it can be used in conjunction with any model including those that are domain specific. However, since our work primarily focuses on studying the expressivity of positional/structural encodings, our evaluation is geared towards comparing MoSE to existing encoding methods. We think this is crucial for a fair comparison across different encoding techniques.
>
> >Regarding the QM9 experiment, I am computer scientist myself and cannot entirely judge the validity, but wanted to point out that some chemists have strong doubts about this kind of experiment:
>
> **Answer:** As stated previously, our experiments are designed to provide a fair comparison of MoSE to other encoding techniques. Therefore, we default to common benchmarks within the GNN and graph community, which include the QM9 dataset. However, we acknowledge that this dataset may not be perfect and provide further experiments on additional datasets in our global response.
>
> >Maybe the authors can point out why they evaluate on only these datasets.
>
> **Answer:**  We originally focused on ZINC and QM9 due to the computational constraints of Graph Transformer (GT) models. Since GTs scale quadratically with the number of nodes in a graph, they are extremely resource intensive, which is why most current GT benchmarks are on molecular datasets.
>
> Additionally, QM9 has 13 different tasks to regress over, whereas most other datasets only have one of two. Therefore, we believe that it is a much more robust dataset to evaluate against and it highlights the flexibility of MoSE to predict a range of properties.
>
> However, we also provide additional experimental results in our global response which further show the performance of MoSE across different types of molecular datasets (PCQM4Mv2, peptides-func, peptides-struct) and different domains (CIFAR10, MNIST).

---

> > ### Author Response · Authors · 2024-11-25
> >
> > We kindly remind the reviewer that the discussion period will be ending soon. Therefore, we would greatly appreciate it if you could share any outstanding questions.

---

> ### Comment · Reviewer_7PEY · 2024-11-25
>
> I've read through the rebuttals and the additional results are good for verification. Overall, I think my rating still reflects my overall impression of the submission.
>
> I slightly disagree with the authors in that I think, if the paper focuses on chemistry datasets, then the evaluation should also be reasonable in this respect (but this doesn't need further discussion).

---

### Official Review · Reviewer_bcpS · 2024-11-02

**Soundness:** 4
**Presentation:** 4
**Contribution:** 3
**Rating:** 8
**Confidence:** 4

**Summary:**

This paper studies the topic of positional/structural encodings for graph learning which are typically influential in Graph Transformers since these incorporate the graph inductive biases coming from the adjacency matrix as its often not directly used during the self attention computation. The paper investigates the expressivity of one of such encodings, the random walk based structural encoding (RWSE), and address their shortcomings by proposing a homomorphism count based structural encoding called MoSE, which are claimed powerful than the RWSE. The content of the paper presents the characteristics brought by MoSE, also using example illustrations where RWSE may fail, and demonstrates in details their performance advantages on molecular datasets as well as one synthetic dataset of random graphs.

**Strengths:**

- the paper is a beautiful read on the limitations of existing RW structural encoding for graph transformers, and how the proposed homomorphism count based SE can address it.
- although the paper does not present a sophisticated new model, the presented SE is carefully studied, and experiments show the advantage (although marginal) brought by MoSE over existing methods in GTs.
- the comparison section of RWSE and MoSE is insightful and shows RWSE is strictly weaker than MoSE, which is reflected in the experimental results, as well as one of the ZINC illustrations.
- the proposed structural encoding has no limitations of LapPE such as sign ambiguity

**Weaknesses:**

- although the MoSE is useful as studied, a concerning factor is of the computational complexity of getting the vectors. how does this compare experimentally?
- the utility of the propose structural encoding may be limited to addressing specific failure cases of prior SEs as indicated by the minor changes in experimental numbers.

**Questions:**

included with weaknesses.

---

> ### Author Response · Authors · 2024-11-21
>
> We sincerely thank Reviewer bcps for the review and positive feedback. We address your comments below:
> >although the MoSE is useful as studied, a concerning factor is of the computational complexity of getting the vectors. how does this compare experimentally?
>
> **Answer:** Computing the homomorphism counts for MoSE is highly parallelizable since the homomorphism from each pattern can be counted independently from one another. Empirically, it took only **~0.02 sec** to compute MoSE for a given graph, whereas training various Transformer models on datasets such as QM9 took as long as **15-30 hours** for a single run. Therefore, the time to compute MoSE is almost negligible overall.
>
> >the utility of the propose structural encoding may be limited to addressing specific failure cases of prior SEs as indicated by the minor changes in experimental numbers.
>
> **Answer:** We provide an additional experiment where we evaluate the self-attention Graph Transformer (Dwivedi & Bresson, 2020) on the peptides-struct and peptides-func datasets to isolate the performance gain of MoSE vs RWSE. The empirical results support the benefits of MoSE which leads to substantial improvements over RWSE (please see our global response for the specific results).
>
> We also extend the ablation study of positional/structural encodings from GPS (Rampasek et al., 2022) with the CIFAR10 dataset, which again highlights the benefits of MoSE across different domains.
>
>
> References:
>
> Dwivedi & Bresson. A generalization of transformer networks to graphs. arXiv preprint arXiv:2012.09699, 2020.
>
> Rampasek et al. Recipe for a general, powerful, scalable graph transformer. In *NeurIPS*, 2022.

---

> > ### Author Response · Authors · 2024-11-25
> >
> > We once again thank the reviewer for their support and positive review. If there is any additional clarification that we can provide, please do not hesitate to let us know.

---

> > > ### Comment · Reviewer_bcpS · 2024-11-26
> > >
> > > Thank you for the response. I acknowledge the answers to my two comments in the original review, including the clarification of time which can be negligible with respect to the total training time, as well as the additional fairer comparison of MoSE vs RWSE. This work overall has a good contribution.

---

### Official Review · Reviewer_yaCD · 2024-11-04

**Soundness:** 2
**Presentation:** 3
**Contribution:** 3
**Rating:** 8
**Confidence:** 3

**Summary:**

The paper introduces MoSE, a graph structural encoding based on homomorphism counts. The authors present theoretical arguments suggesting that MoSE has greater expressive power than RWSE and demonstrate competitive empirical performance when integrated with GTs.

**Strengths:**

1. The experiments supports the claim that MoSE improves performance on molecular property prediction tasks.
2. The theoretical arguments provide a plausible explanation for the observed performance gains.
3. The paper is well-organized and easy to follow.

**Weaknesses:**

1. The paper focuses on comparing MoSE with RWSE, justified by RWSE's empirical success. However, several recent graph transformers, such as GRIT, can incorporate edge features and encodings. RRWP, as proposed in the GRIT paper, builds on RWSE by incorporating non-diagonal terms and achieves both greater theoretical expressiveness and empirical performance improvements over RWSE. Although the paper shows that GRIT+MoSE outperforms GRIT+RRWP, a theoretical explanation (e.g., specific examples or proofs, similar to the RWSE discussion) would strengthen this claim. Additionally, more detailed discussion on when and why node-level structural encodings remain competitive despite SOTA GNNs' ability to process edge features and encodings would be beneficial.
2. MoSE's evaluation is based on two molecular datasets, ZINC and QM9. While these datasets are standard in GNN benchmarking, limiting evaluation to these two datasets might be too restrictive.
    - If the paper's primary aim is to propose an encoding specifically for molecular property prediction (as supported by Example 1, which is plausible), evaluation across additional molecular datasets, such as [PCQM4M](https://pytorch-geometric.readthedocs.io/en/2.5.2/generated/torch_geometric.datasets.PCQM4Mv2.html#torch_geometric.datasets.PCQM4Mv2), [Peptides-struct](https://pytorch-geometric.readthedocs.io/en/2.5.2/generated/torch_geometric.datasets.LRGBDataset.html#torch_geometric.datasets.LRGBDataset), [Peptides-func](https://pytorch-geometric.readthedocs.io/en/2.5.2/generated/torch_geometric.datasets.LRGBDataset.html#torch_geometric.datasets.LRGBDataset), and [AQSOL](https://pytorch-geometric.readthedocs.io/en/2.5.2/generated/torch_geometric.datasets.AQSOL.html#torch_geometric.datasets.AQSOL), could reinforce the validity of MoSE within this domain. Furthermore, clarifying this scope within the paper—by showing that the phenomena in Example 1 generalize across molecular property tasks—would align well with the empirical objectives. My overall rating assumes this as the paper’s main focus.
    - Conversely, if the goal is to propose a general-purpose graph encoding, then broader evaluation across datasets from other domains is necessary. Evaluating MoSE beyond molecular property datasets would better support a general claim. In this case, datasets from [GNN Benchmark Datasets](https://pytorch-geometric.readthedocs.io/en/2.4.0/generated/torch_geometric.datasets.GNNBenchmarkDataset.html) beyond ZINC can help validate MoSE's general applicability.

**Questions:**

What are the time and space complexities for computing MoSE encodings? How does the empirical preprocessing time compare to RWSE, RRWP, and LapPE?

---

> ### Author Response · Authors · 2024-11-21
>
> Thank you for your review. We have addressed your questions and concerns here:
> >Although the paper shows that GRIT+MoSE outperforms GRIT+RRWP, a theoretical explanation (e.g., specific examples or proofs, similar to the RWSE discussion) would strengthen this claim
>
> **Answer:** This is a good point, but unfortunately, it is difficult to directly compare encodings on individual nodes to encodings for node pairs. That being said, one can extend MoSE encodings naturally to pairs of vertices (by fixing two vertices in the pattern graphs). In that setting, our proofs that show that RWSE is simply a special case of MoSE also easily adapt to showing that RRWP is a special case of pair-wise MoSE (the patterns being paths with the same weighting scheme as for RWSE).
>
> >Additionally, more detailed discussion on when and why node-level structural encodings remain competitive despite SOTA GNNs' ability to process edge features and encodings would be beneficial.
>
> **Answer:** Node-level structural encodings may sometimes provide more explicit information about graph structure than what can be captured by some GNNs (e.g., cycles cannot be detected by GNNs). Furthermore, in order to learn longer-range information, GNNs must go through several layers of message-passing, which may distort the graph signal and lead to information loss due to over-squashing (Alon & Yahav, 2021).
>
> >MoSE's evaluation is based on two molecular datasets, ZINC and QM9. While these datasets are standard in GNN benchmarking, limiting evaluation to these two datasets might be too restrictive.
>
> **Answer:** We thank the reviewer for their suggestion and provide additional experimental results in our global response. These further show the performance of MoSE for additional molecular datasets (PCQM4Mv2, peptides-func, peptides-struct) and across different domains (CIFAR10, MNIST).
>
> >What are the time and space complexities for computing MoSE encodings? How does the empirical preprocessing time compare to RWSE, RRWP, and LapPE?
>
> **Answer:** From a theoretical perspective, the complexity of MoSE is very intricate. With constant uniform weighting, counting homomorphisms from H to G is feasible in time $O(|G|^{tw(H)})$ where $tw(H)$ is the treewidth of H. It is known that this exponent cannot be substantially improved upon (Marx, 2007). The complexity of computing MoSE encodings thus fundamentally depends on the choice of patterns and their treewidth.
>
> As we show in Proposition 5, RWSE is simply a limited special case of MoSE for one specific set of pattern graphs (cycles) and one specific weighting scheme ($w(v) = 1/degree(v)$). Cycles always have treewidth 2, and thus RWSE is always feasible in quadratic time.
>
> Since MoSE generalizes popular encodings, we see it also as a way to trade off more compute to achieve a more informative (and thus ideally better performing and generalizing) encoding. This kind of adaptability is not possible in RWSE or other popular encodings.
>
> For further details on the empirical processing time of MoSE and how its complexity compares to other encoding methods, please refer to our global response.
>
>
> References:
>
> Alon & Yahav. On the Bottleneck of Graph Neural Networks and Its Practical Implications. In *ICLR*, 2021.
>
> Marx, D. Can you beat treewidth? In *FOCS*, 2007.

---

> > ### Comment · Reviewer_yaCD · 2024-11-21
> >
> > I would like to thank the authors for their detailed response. The new experimental results look promising, and adress my main concern.
> >
> > In the additional experiments, why do the authors use GT as the baseline model on peptides-func and peptides-struct, while using GPS on other datasets? Can the authors further justify this choice?

---

> > > ### Author Response · Authors · 2024-11-21
> > >
> > > Thank you for acknowledging our rebuttal - we are happy that your main concern is addressed. Regarding GT vs GPS, the main difference is that the latter uses MPNN layers, which already provides a graph inductive bias. Since we are interested in mainly identifying the graph inductive bias provided by positional encodings, we found it meaningful to conduct some experiments with GT to eliminate confounding factors as much as possible.
> > >
> > > Furthermore, we think it is valuable to experiment relative to a wide range of graph transformer variations on different datasets to show that the overall trends are essentially similar. For the final version of the paper, we can report results uniformly relative to one model, i.e., GT  and defer some of the results concerning model variations to the appendix.

---

> > > > ### Comment · Reviewer_yaCD · 2024-11-22
> > > >
> > > > This is a valid explanation. I have changed my score to reflect the fact that my concerns are addressed.

---

> > > > > ### Author Response · Authors · 2024-11-22
> > > > >
> > > > > We thank the reviewer for all their feedback and for raising their score based on our rebuttal.

---

### Official Review · Reviewer_MzSj · 2024-11-04

**Soundness:** 3
**Presentation:** 3
**Contribution:** 2
**Rating:** 6
**Confidence:** 3

**Summary:**

The paper proposes Motif Structural Encoding (MoSE), a novel structural encoding method based on homomorphism counts for graph learning tasks. MoSE leverages homomorphism counts to represent graph structure effectively, aiming to surpass limitations found in common positional and structural encodings such as Random Walk Structural Encoding (RWSE). Theoretically, MoSE demonstrates enhanced expressiveness compared to RWSE and is shown to be independent of the Weisfeiler-Leman (WL) hierarchy, thus providing a broader graph representation power. The paper provides empirical validation, showing that MoSE achieves superior results over other encoding methods on various datasets, including molecular property prediction on ZINC and QM9.

**Strengths:**

- MoSE’s homomorphism count-based encoding enhances expressiveness beyond RWSE and aligns well with complex graph structures, as it provides unique structural insights through flexible motif selection.
- The paper rigorously compares MoSE to established methods and relates its expressiveness to the WL hierarchy, grounding its effectiveness in theoretical proofs.
- MoSE achieves state-of-the-art results on benchmarks, showcasing its applicability and effectiveness in both real-world and synthetic datasets, particularly in molecular graph prediction tasks.

**Weaknesses:**

- The performance of MoSE may vary depending on the choice of motif graphs, requiring task-specific tuning to achieve optimal results.
- Calculating homomorphism counts for large and complex graphs could increase computational requirements, potentially limiting scalability for very large datasets.
- The theoretical analysis of the expressivity of homomorphism counts has been established and well-studied by recent works (e.g., Jin et al. 2024). It is not very clear what additional theoretical contribution this paper has.

**Questions:**

Please see Weaknesses

---

> ### Author Response · Authors · 2024-11-21
>
> Thank you for your constructive feedback. We respond to your main points below:
>
> >The performance of MoSE may vary depending on the choice of motif graphs, requiring task-specific tuning to achieve optimal results.
>
> **Answer:** The ability to customize MoSE to specific tasks is actually a benefit, since encodings such as RWSE are not able to accommodate such flexibility. Because MoSE **strictly generalizes** RWSE, the performance of MoSE will at the very least be equal to that of RWSE, with task-specific tuning only increasing the performance of MoSE.
>
> If there is no obvious choice of motifs, we can construct MoSE using the set of all connected graphs up to $k$ vertices, which would allow us to capture a wide range of graph motif parameters. We show an example of this in our QM9 experiment, where it is unclear which motifs will help with which of the 13 properties. Therefore, MoSE is constructed using the homomorphism counts for the set of all connected graphs up to 5 vertices and the 6-cycle.
>
> >Calculating homomorphism counts for large and complex graphs could increase computational requirements, potentially limiting scalability for very large datasets.
>
> **Answer:** In practice, for small $k$ (=size of the motifs), the bottleneck in training is due to the quadratic runtime complexity of Graph Transformers rather than the computation of the homomorphism counts. For further details on the complexity of MoSE, please also refer to our global response.
>
> >The theoretical analysis of the expressivity of homomorphism counts has been established and well-studied by recent works (e.g., Jin et al. 2024). It is not very clear what additional theoretical contribution this paper has.
>
> **Answer:** Our work specifically studies the power of homomorphism counts as a graph inductive bias in their own right, and we prove that MoSE strictly generalizes RWSE, which is a very popular encoding technique. Although our work is closely related to Jin et al. 2024, their focus is on identifying which motif parameters should be used to reach a desired level of expressivity, whereas our focus is to study the expressivity of the homomorphism counts as a structural encoding technique themselves in comparison with existing encodings.
>
> References:
>
> Jin et al. Homomorphism Counts for Graph Neural Networks: All About That Basis. In *ICML*, 2024.

---

> > ### Author Response · Authors · 2024-11-25
> >
> > We kindly remind the reviewer that the discussion period will be ending soon. Therefore, we would greatly appreciate it if you could share any outstanding questions.

---

> ### Comment · Reviewer_MzSj · 2024-11-26
>
> I appreciate the authors’ response which has addressed some of my concerns. Still, I don’t think using homomorphism counts as PE brings significant contributions given existing works on the expressivity of homomorphism counts. It would be more interesting if the transformer architecture itself could play a role in the analysis, while now it seems to be more of a restatement of established graph theory. But overall it’s a good paper and I recommend acceptance.

---

### Author Response · Authors · 2024-11-21
**Global Response**

We thank the reviewers for their comments and address each remark in detail in our individual responses. We also include further experimental results and complexity details here.

**Additional experimental results:**
1. We extend the comparison of different positional/structural encodings from GPS (Rampasek et al., 2022) with the PCQM4Mv2-subset and CIFAR10 datasets:


| Encoding     | PCQM4Mv2-subset (MAE $\downarrow$)| CIFAR10 (Acc $\uparrow$)|
| --------     | :--------: | :--------: |
| *none*       | 0.1355   | 71.49    |
| PEG-LapEig   | 0.1209   | 72.10    |
| LapPE        | 0.1201   | 72.31    |
| SignNet-MLP  | 0.1158   | 71.74    |
| SignNet-DeepSets |0.1144| 72.37    |
| RWSE         | 0.1159   | 71.96    |
| **MoSE (ours)**  | **0.1133** | **73.50**    |

2. We evaluate the self-attention Graph Transformer (Dwivedi & Bresson, 2020) on the peptides-struct and peptides-func datasets to isolate the performance gain of MoSE vs RWSE:

|          | Peptides-struct (MAE $\downarrow$) | Peptides-func (AP $\uparrow$) |
| -------- | :--------: | :--------: |
| GT+RWSE  | $0.344 \pm \tiny0.009$     | $62.52 \pm \tiny1.15$     |
| **GT+MoSE (ours)** | $\mathbf{0.318 \pm \tiny0.010}$ | $\mathbf{63.49 \pm \tiny1.13}$ |


3. Due to the encouraging results from GPS+MoSE on the CIFAR10 image dataset, we ran an additional experiment on MNIST, which is another standard GNN dataset:


| Model    | MNIST (Acc $\uparrow$) |
| -------- | :--------: |
| EGT      | $98.173 \pm \tiny0.087$     |
| GPS+LaPE | $98.051 \pm \tiny0.126$     |
| GRIT     | $98.108 \pm \tiny0.111$     |
| **GPS+MoSE (ours)** | $\mathbf{98.198 \pm \tiny0.002}$ |


These additional experiments further validate the benefit of using MoSE and support the  theoretical findings of our paper. We thank the reviewers for their pointers which helped us to improve the paper.



**Complexity of computing MoSE encodings:**

Practically, computing the homomorphism counts for MoSE is highly parallelizable since the homomorphisms from each pattern can be counted independently from one another. In our experiments, computing the MoSE encodings took only **~0.02 sec** per graph. By contrast, training various Transformer models on datasets such as QM9 took between **15-30 hours** for a single run. Therefore, the time to compute MoSE is almost negligible overall.

From a theoretical perspective, the complexity of MoSE is very intricate. With constant uniform weighting, counting homomorphisms from H to G is feasible in time $O(|G|^{tw(H)})$ where $tw(H)$ is the treewidth of H. It is known that this exponent cannot be substantially improved upon (Marx, 2007). The complexity of computing MoSE encodings thus fundamentally depends on the choice of patterns and their treewidth.

As we show in Proposition 5, RWSE is simply a limited special case of MoSE for one specific set of pattern graphs (cycles) and one specific weighting scheme ($w(v) = 1/degree(v)$). Cycles always have treewidth 2, and thus RWSE is always feasible in quadratic time. However, RWSE cannot capture other treewidth 2 motifs that MoSE can accommodate for essentially the same cost.

We will include these experiments and an explicit discussion on complexity in the revised version of our paper, and thank the reviewers again for these very helpful suggestions.


References:

Rampasek et al. Recipe for a general, powerful, scalable graph transformer. In *NeurIPS*, 2022.

Marx, D. Can you beat treewidth? In *FOCS*, 2007.

---

### Meta-Review · Area_Chair_xak4 · 2024-12-22

**Metareview:**

(a) The paper proposes Motif Structural Encoding (MoSE), a novel structural encoding method for graph learning. It shows MoSE has greater expressive power than Random Walk Structural Encoding (RWSE) and is independent of the Weisfeiler-Lehman (WL) hierarchy. Empirically, MoSE outperforms other encodings on various datasets, especially in molecular property prediction.

(b) Strengths:
Theoretically, MoSE's expressiveness is rigorously analyzed and compared with RWSE, providing a solid theoretical foundation.
Empirically, it achieves state-of-the-art results on benchmarks, validating its effectiveness.
The paper is well-structured and the presentation is clear, facilitating understanding.

(c) Weaknesses:
MoSE's performance may depend on motif graph choice, requiring task-specific tuning.
The theoretical contribution regarding homomorphism counts' expressivity may not be highly novel compared to some recent works.

(d) Reasons for acceptance:
Despite the weaknesses, the paper makes a significant contribution. The proposed MoSE is a powerful encoding technique with clear theoretical and empirical advantages over existing methods. The authors addressed reviewers' concerns well, providing additional experiments and explanations. For example, they showed the negligible computational time for MoSE compared to training time, and demonstrated its flexibility in different scenarios. The overall quality of the paper, in terms of scientific content and presentation, justifies its acceptance.

**Additional Comments On Reviewer Discussion:**

Reviewer points and author responses:

- Performance variation with motif graphs: Reviewers questioned the impact of motif graph choice on MoSE's performance. Authors responded that MoSE's flexibility is an advantage, and in the absence of obvious motif choices, they can use a set of all connected graphs  up to k vertices. They also provided an example in the QM9 experiment.
- Computational complexity: Concerns were raised about the computational requirements of MoSE. Authors explained that in practice, the time to compute MoSE is almost negligible compared to the training time of Graph Transformers. They also discussed the theoretical complexity and its relationship with pattern choice and treewidth.
- Theoretical contribution: Reviewers doubted the novelty of the theoretical analysis. Authors clarified that their focus is on studying the power of homomorphism counts as a graph inductive bias in comparison with existing encodings, which is different from related works.
- Comparison with other graph transformers: Reviewers asked for a theoretical explanation of MoSE's performance compared to other graph transformers like GRIT+RRWP. Authors showed that RRWP can be seen as a special case of pairwise MoSE.
- Evaluation scope: Reviewers suggested expanding the evaluation to more datasets. Authors provided additional experimental results on various datasets, including PCQM4Mv2, peptides-func, peptides-struct, CIFAR10, and MNIST.
- Model choice in experiments: Reviewers questioned the choice of baseline models in different experiments. Authors explained that they used different models to isolate the graph inductive bias provided by positional encodings and to show overall trends.

Weighing:

The authors' responses addressed most of the reviewers' concerns effectively. The additional experiments and clarifications strengthened the paper. The negligible computational time for MoSE and its demonstrated flexibility in motif choice were important factors. The theoretical explanations, although not completely novel, still provided valuable insights. The expansion of the evaluation to more datasets also enhanced the paper's comprehensiveness. Overall, these factors outweighed the remaining weaknesses and supported the decision to accept the paper.

---

### Decision · Program_Chairs · 2025-01-22

Accept (Poster)